# *Bacteroides Fragilis* in the gut microbiomes of Alzheimer's disease activates microglia and triggers pathogenesis in neuronal C/EBPβ transgenic mice

Yiyuan Xia [1,2,10], Yifan Xiao[2,10], Zhi-Hao Wang[1], Xia Liu[1], Ashfaqul M. Alam[3], John P. Haran[4,5,6], Beth A. McCormick[5,6], Xiji Shu[2] ✉, Xiaochuan Wang [7,8] ✉ & Keqiang Ye [1,9] ✉

Gut dysbiosis contributes to Alzheimer's disease (AD) pathogenesis, and Bacteroides strains are selectively elevated in AD gut microbiota. However, it remains unknown which Bacteroides species and how their metabolites trigger AD pathologies. Here we show that *Bacteroides fragilis* and their metabolites 12-hydroxy-heptadecatrienoic acid (12-HHTrE) and Prostaglandin E2 (PGE2) activate microglia and induce AD pathogenesis in neuronal C/EBPβ transgenic mice. Recolonization of antibiotics cocktail-pretreated Thy1-C/EBPβ transgenic mice with AD patient fecal samples elicits AD pathologies, associated with C/EBPβ/Asparaginyl endopeptidase (AEP) pathway upregulation, microglia activation, and cognitive disorders compared to mice receiving healthy donors' fecal microbiota transplantation (FMT). Microbial 16S rRNA sequencing analysis shows higher abundance of proinflammatory *Bacteroides fragilis* in AD-FMT mice. Active components characterization from the sera and brains of the transplanted mice revealed that both 12-HHTrE and PGE2 activate primary microglia, fitting with poly-unsaturated fatty acid (PUFA) metabolites enrichment identified by metabolomics. Strikingly, recolonization with live but not dead *Bacteroides fragilis* elicited AD pathologies in Thy1-C/EBPβ transgenic mice, so did 12-HHTrE or PGE2 treatment alone. Collectively, our findings support a causal role for *Bacteroides fragilis* and the PUFA metabolites in activating microglia and inducing AD pathologies in Thy1- C/EBPβ transgenic mice.

The neuroinflammation, triggered by insoluble amyloid β (Aβ) peptide deposits and phosphorylated Tau-enriched neurofibrillary tangles (NFT) that are two prominent hallmarks in Alzheimer's disease (AD), significantly contributes to AD pathogenesis[1]. The innate immune cells involved in this process are primarily microglia and astrocytes[2]. Activated microglia play a key role in the brain's immune response to neuronal degeneration, and microglial activation is an early event in the pathogenesis of AD[3]. The gut microbiome is essential for microglial functions in immune and neuronal responses throughout the host's life span. Host microbiota substantially contributes to microglia homeostasis, and microglia in germ-free mice display global defects including altered cell proportions, and immature phenotype

associated with impaired innate immune response. The absence of a complex host microbiota leads to defects in microglia maturation, differentiation, and function[4]. Gut microbiota imbalance leads to increased permeability of the intestinal epithelial barrier with the release of proinflammatory cytokines and promotion of a neuroinflammatory response[5]. Aging-induced changes in gut microbiota compositions modulate the morphology and functions of microglia through the gut-brain axis. Gut microbiota communicates with microglia by its secreted metabolites and neurotransmitters. This is highly associated with age-related cognitive declines[6]. It is worth noting that increased proinflammatory and reduced anti-inflammatory bacteria in the intestine are associated with systemic inflammatory states in patients with cognitive impairment and brain amyloidosis[7]. Recent studies revealed significant changes in the proportion of certain microbiome taxa in AD patients[8,9]. Notably, AD patient microbiomes are associated with dysregulation of the anti-inflammatory P-glycoprotein pathway, revealing a lower proportion and prevalence of bacteria with the potential to synthesize butyrate, as well as higher abundances of taxa that cause inflammation[10]. Most recently, we showed that AD-derived microbiota enhances the proinflammatory pathway for poly-unsaturated fatty acid (PUFA) metabolism, and Bacteroides strains mediating PUFA metabolism are increased in AD patient gut microbiomes, activating microglia in the brain, associated with C/EBPβ/AEP signaling activation[11].

Recently, we identified δ-secretase, an asparagine endopeptidase (AEP, gene name *LGMN*), which cleaves both APP and Tau in an age-dependent manner in the brains, where its enzymatic activity and expression level are temporally escalated. It cuts APP at both N373 and N585 residues and facilitates Aβ production by decreasing the steric hindrance for BACE1[12]. Additionally, δ-secretase cuts Tau at N255 and N368 and abolishes its microtubule assembly activity, resulting in its aggregation and NFT formation. Depletion of δ-secretase significantly reduces Aβ production and NFT formation in AD mouse brains, leading to restoration of synaptic activity and cognitive functions[13]. Hence, AEP is necessary for AD onset and progression. C/EBPβ, an Aβ and IL-6-activated transcription factor[14], regulates genes critical for the activation of microglia, and it also mediates numerous cytokines and pro-inflammatory gene expression[15,16]. Remarkably, C/EBPβ is highly upregulated and activated in AD brains[17–19], modulating Tau pathology propagation in microglia[20]. We show that C/EBPβ is highly expressed in human AD brains, especially in neurons, inducing AD pathologies by upregulating AEP[19]. Moreover, we report that spatiotemporal activation of the C/EBPβ/AEP axis regulates the pathogenesis of AD[21]. In addition, we disclose that C/EBPβ preferentially dictates ApoE4 mRNA transcription in AD brains[22], which feeds back and activates C/EBPβ in the presence of 27-hydroxycholesterol, exacerbating AD pathologies[23]. Most recently, we report that C/EBPβ is escalated in neurons in an age-dependent manner and mediates *APP*, *MAPT*, and *BACE1* mRNA expression, and mice with double transgenic expression of ApoE4 and C/EBPβ in neurons exhibit sporadic AD pathologies in Thy1-ApoE4/C/EBPβ mouse model[24]. Thus, C/EBPβ/AEP pathway plays a critical role in driving AD pathogenesis. Further, we report that FSH (follicle stimulatory hormone) selectively activates C/EBPβ/AEP signaling after menopause, leading to women's more susceptibility to AD onset[25].

In the central nervous system (CNS), cyclooxygenases (COXs) and 5-lipoxygenase (LOX-5), enzymes that metabolize arachidonic acid (AA) into biologically active lipid molecules termed eicosanoids, are associated with pathobiological mechanisms accompanying aging and neurodegeneration[26–28]. However, both COXs and LOX-5 also influence CNS functions via mechanisms unrelated to the roles in inflammation. For example, COX-2, which is predominantly expressed in pyramidal neurons in contrast to COX-1, which is mostly present in microglia, regulates neuroplasticity via its conversion of AA to classic prostaglandins but also by favoring oxidative metabolism of endo-cannabinoids to novel prostaglandins[29]. Noticeably, AD patient brains contain more AA than normal healthy controls[30]. A portion of intra-cellular free AA is metabolized by COX-1, COX-2, and LOX-5 to biologically active prostaglandins and leukotrienes, respectively. COX-2 and LOX-5 single nucleotide polymorphisms (SNP) study suggests that the alleles of COX-2 and LOX-5 could be risk factors for AD[31]. Interestingly, LOX-5 regulates Aβ levels in the brain by influencing γ-secretase[32].

In the present study, we show that AD-FMT induced AD-like pathologies and cognitive impairments in Thy1-C/EBPβ transgenic mice mimic augmented C/EBPβ levels in neurons in aged animals. Moreover, we identify *Bacteroides fragilis* is selectively enriched in the gut microbiota, and these bacterial PUFA metabolites 12-HHTrE or PGE2 trigger primary microglia activation. Remarkably, *Bacteroides fragilis* or 12-HHTrE or PGE2 alone induces AD-like pathologies and cognitive disorders by activating C/EBPβ and microglia in the brain of Thy1-C/EBPβ transgenic mice.

## Results

### AD patient fecal transplantation induces mouse Aβ and Tau aggregations in Thy1-C/EBPβ transgenic mice

C/EBPβ expression is escalated in neurons in an age-dependent manner[33]. Interestingly, neuronal C/EBPβ represses *REST/FOXO* mRNA transcription, shortening the lifespan of Thy1-C/EBPβ transgenic mice[34]. AD gut microbiota stimulates C/EBPβ/AEP pathway and enhances PUFA metabolism that activates microglia in the brain of germ-free 3xTg mice[11]. To explore whether AD gut dysbiosis is capable to trigger AD-like pathologies using mouse machinery, we pretreated Thy1-C/EBPβ transgenic mice with antibiotics cocktail (Abx) for 1 month, and fecal pellets in vitro culture suggested that Abx treatment substantially depleted the host gut microbiota (Supplementary Fig. 1A, B). Mice were orally gavaged once a day consecutively for 2 months with frozen fecal samples obtained from 4 donors (2 AD and 2 HC (healthy control), 7–8 mice per donor). Then, mouse Aβ and Tau pathologies were analyzed via immuno-fluorescent (IF) staining. AD but not HC fecal samples triggered robust Aβ IF signals, which were also Thioflavin S (Th-S) positive, indicating that mouse Aβ is aggregated via β-sheet conformation, resembling pathological Aβ inclusions in the senile plaques in human AD brains. By contrast, there were no signals in wild-type (WT) mice regardless of AD or HC fecal treatment (Fig. 1A). We made a similar observation with aberrant Tau accumulation, which was validated by T22, an aggregated Tau specific antibody (Fig. 1B). Quantification showed that Aβ and Tau aggregates were significantly higher in C/EBPβ transgenic mice than in WT mice (Fig. 1C, D). ELISA assays indicated that mouse Aβ$_{42}$ but not Aβ$_{40}$ peptide concentrations were increased by AD fecal inoculation with C/EBPβ transgenic mice much stronger than WT mice (Fig. 1E). In alignment with Aβ$_{42}$ levels escalation by AD-FMT, immunoblotting (IB) showed that much stronger Aβ aggregates smear in the brain of C/EBPβ transgenic mice, similar to human AD brain lysates (Fig. 1F, left panel), suggestive of demonstrable basal levels of Aβ$_{40}$ peptides in WT mice, which was not elevated in C/EBPβ transgenic mice by HC-FMT. Remarkably, T22 positive p-Tau aggregates were only observed in the brain from C/EBPβ transgenic mice, reminiscent of accumulated Tau in human AD brains (Fig. 1F, right panel). To define that mouse Aβ and Tau were indeed aggregated into the pathological inclusions in the brain, we conducted the immuno-EM (electron microscopy) analysis and found that AD fecal inoculation triggered both mouse Aβ and Tau accumulation (arrow) in WT mouse brain, and the immuno-activities were profoundly augmented in C/EBPβ transgenic mice (Fig. 1G), fitting with the IB findings. Silver staining demonstrated that prominent protein aggregates in the WT brain after AD fecal inoculation, which was further exacerbated in C/EBPβ transgenic mice (Supplementary Fig. 1C, D).

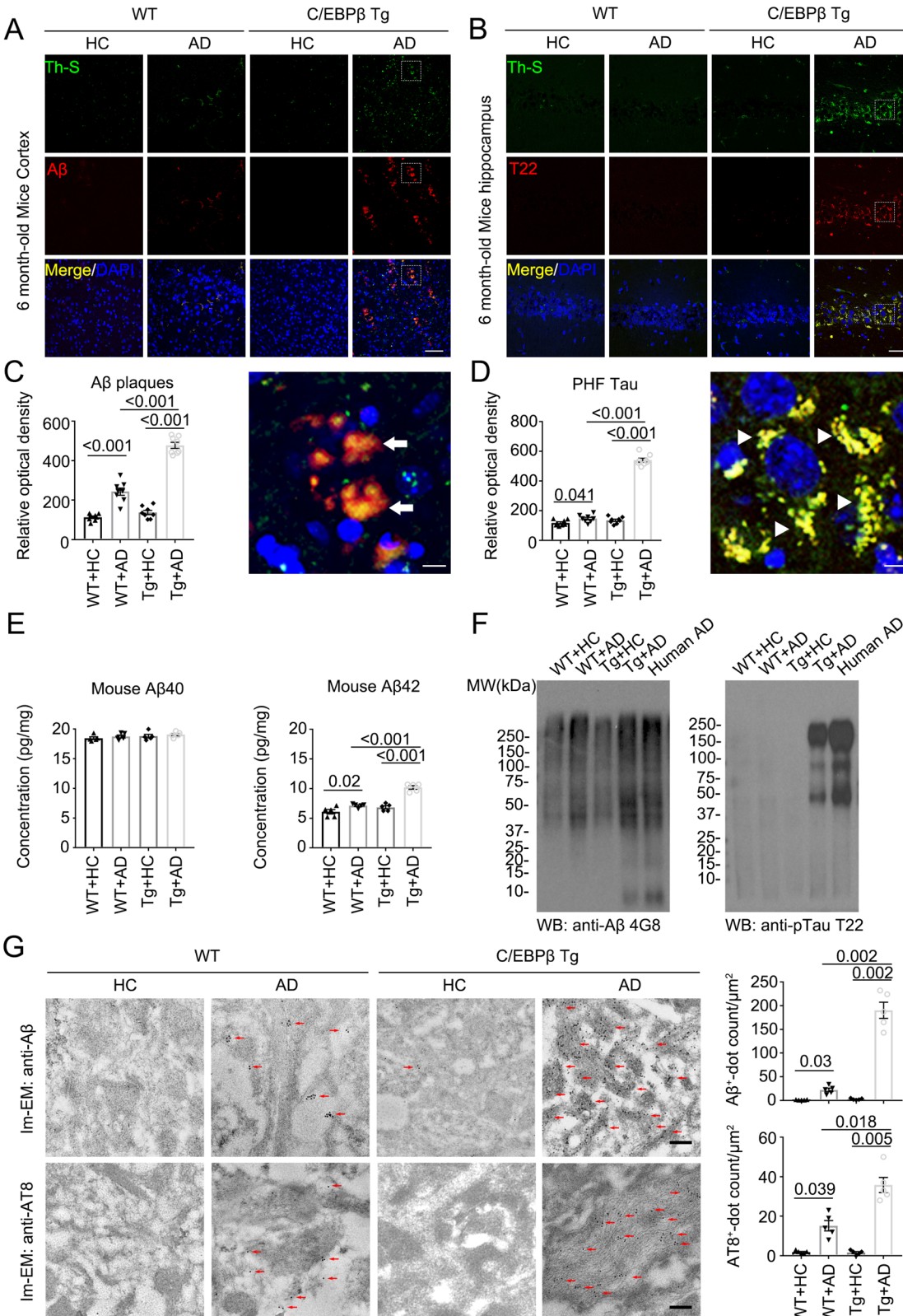

### AD patient fecal transplantation activates C/EBPβ/AEP pathway and cognitive dysfunctions in Thy1-C/EBPβ transgenic mice

To assess whether AD gut microbiota incurred AD-like pathologies implicated C/EBPβ/AEP signaling activation, we performed IB analysis and found that both C/EBPβ and p-C/EBPβ, a biomarker for its activation[35], were elevated in WT mice upon AD fecal inoculation, which subsequently elicited AEP upregulation and activation.

Consequently, APP N585 and Tau N368 fragmentation, the downstream targets of AEP, was evidently increased as compared to HC control. Accordingly, p-Tau AT8 activities were elevated. These biochemical events were pronouncedly intensified in C/EBPβ transgenic mice after AD gut microbiota inoculation (Fig. 2A, B). As expected, the dendritic spines revealed by Golgi staining were reduced in WT mice by AD gut microbiota, which was further decreased in C/EBPβ

**Fig. 1 | AD humanized Abx-treated mice display increased AD pathologies compared with HC humanized Abx-treated mice in WT and C/EBPβ Tg mice.**
**A**, **B** Immunofluorescent staining of Aβ (red) and Th-S (green) in the frontal cortex region of brains, Th-S (green) and T22 (red) in the hippocampus CA1 region of brains from 6-month-old AD humanized Abx-treated mice and HC humanized Abx-treated mice. Scale bar: 40 μm. **C** Quantitative analysis of Aβ and Th-S positive plaques (left panel, *n* = 8 biologically independent samples in each group, data are shown as mean ± SEM. one-way ANOVA and Bonferroni's multiple comparison test). A high magnification picture showed Aβ plaques (Arrow) were significantly increased in AD humanized Abx-treated mice brain (right panel, Scale bar: 10 μm). **D** Quantitative analysis of T22 and Th-S positive PHFs (left panel, n = 8 biologically independent samples in each group, data are shown as mean ± SEM. one-way ANOVA and Bonferroni's multiple comparison test). A high magnification picture showed PHF Tau (Arrowhead) were significantly increased in AD humanized Abx-treated mice brain (right panel, Scale bar: 10 μm). **E** $A\beta_{40}$ and $A\beta_{42}$ concentrations in the cortex of AD or HC humanized Abx-treated mice were measured using mouse $A\beta_{40}$ and $A\beta_{42}$ ELISA kit. the concentration of $A\beta_{42}$, not $A\beta_{40}$ was significantly increased in AD humanized Abx-treated mice compared with HC humanized Abx-treated mice cortex. (n = 5 biologically independent samples in each group, data are shown as mean ± SEM, one-way ANOVA and Bonferroni's multiple comparison test). **F** Enrichment of aggregated Tau and Aβ in the detergent-insoluble fractions of mouse brains and AD human cortex. Sarkosyl-insoluble fractions were blotted with antibodies against Aβ (4G8), or Tau T22 for Tau oligomers (representative of 3 mice). **G** immunogold EM showed the aggregated extracellular Aβ and intracellular AT8 in the hippocampus neurons of AD or HC humanized Abx-treated mice. Fibrils like structures (arrows) were highly increased in AD humanized Abx-treated mice (left panels, Scale bar: 200 nm). Quantitative analysis of Aβ and AT8 positive dots. (Right panel, *n* = 5 biologically independent samples in each group, data are shown as mean ± SEM, one-way ANOVA and Bonferroni's multiple comparison test).

transgenic mice (Fig. 2C, D). EM analysis showed that the synapses in WT mice were attenuated by AD fecal samples, which were substantially mitigated in C/EBPβ transgenic mice (Fig. 2E, F). MWM (Morris Water Maze) behavioral tests demonstrated that the escape latency for WT mice was significantly increased by AD fecal transplantation compared to HC, whereas it remained comparable among C/EBPβ transgenic mice regardless of AD or HC gut microbiota inoculation (Fig. 2G), suggesting that the learning capability for WT mice was impaired by AD gut microbiota, and neuronal overexpression of C/EBPβ exerts more prominent detrimental effect than AD gut dysbiosis on the learning ability. By contrast, the time percentage spent in the target quadrant, an index reflecting the memory functions, was significantly reduced in WT mice by AD microbiota as compared to HC, which was further decreased in C/EBPβ transgenic mice, though the swimming speeds remained similar among the groups (Fig. 2H). Hence, AD microbiota activates C/EBPβ/AEP pathway and triggers synaptic degeneration in Thy1-C/EBPβ transgenic mice, leading to cognitive dysfunctions.

## AD fecal transplantation activates microglia and increases PUFA oxidative enzymes in Thy1-C/EBPβ transgenic mice

C/EBPβ regulates pro-inflammatory genes in microglia and is upregulated in AD. Mounting evidence demonstrates that C/EBPβ orchestrates microglial transcriptional programs that promote inflammation and neuronal cell death[18]. Interestingly, it mediates Tau pathology spreading in microglia in AD[20]. IF co-staining revealed that C/EBPβ was upregulated in Iba-1 positive microglia cells in the hippocampus of WT mice upon AD gut microbiota transplantation, and this effect was further elevated in Thy1- C/EBPβ transgenic mice (Fig. 3A, B). Since dysregulated microglia are intimately involved in AD pathogenesis, we conducted IF to analyze microglia morphology on the brain sections. IF co-staining with antibodies against both Iba-1 and CD86, a biomarker for microglia activation[36], exhibited an enlarged cell body and nucleus in C/EBPβ transgenic mice versus WT mice. Further, highly ramified microglia changed to an ameboid form upon AD gut microbiota. CD86 was strongly escalated in AD microglia by the AD fecal samples (Fig. 3C). The diameter of the cell body, the number of branch points and the total branch length in the microglia were quantified, as the diameters increased, the branch points and length were significantly diminished (Fig. 3D). Correspondingly, the inflammatory cytokines including IL-6 and IL-1β were strongly augmented by AD gut microbiota (Fig. 3E), in accordance with the microglia activation status. IF co-staining using anti-CD86 with Iba-1 or IL-1β, or Iba-1 with IL-1β or IL-6 demonstrated that AD fecal transplantation strongly activated Iba-1 positive microglia from WT mice with CD86 signals significant elevation than HC samples, and this effect was further augmented in microglia from C/EBPβ Tg mice. IL-1β and IL-6 co-staining exhibited the same patterns, and these events were quantified (Supplementary Fig. 2A–D). In alignment with the robust microglia activation, human

$A\beta_{42}$ aggregates were greatly engulfed into the active microglia, resulting in a reduction of remnant $A\beta_{42}$ in the media. As expected, microglia from C/EBPβ Tg mice exhibited stronger activities than WT mice (Supplementary Fig. 3).

Gut microbiota activates C/EBPβ/AEP pathway, elevating proinflammatory enzymes implicated in PUFA metabolism, and PGE2 activates microglia in germ-free 3xTg mice in the presence of SCFA (short chain fatty acid)[11]. IB analysis revealed that AD gut microbiota increased PUFA enzymes including LOX-5; COX-1, COX-2, BLT-2, and PTGES in the brain of WT mice, which were further escalated in Thy1-C/EBPβ transgenic mice (Fig. 3F, G). Quantitative RT-PCR (qRT-PCR) validated the *ALOX5*, *PTGES1*, and *PTGES* but not *PTGS2, LTB4R1*, or *LTB4R2* genes were selectively agitated upon AD fecal treatment (Fig. 3H). These findings are consistent with previous reports that *COX*s and *LOX-5* and *PTGES* mRNA expression is mediated by C/EBPβ[37–39].

Dysregulated microglia are intimately involved in neurodegeneration. Chronic neuro-inflammation mediated by microglia is one of the pathological hallmarks of AD[20], and microglial activation is an early event in the pathogenesis of AD[3]. To search for the pathogens that activated microglia in the brain in the AD fecal inoculated mice, we conducted in vitro primary microglial activation assay using the serum from the mice pretreated with Abx, followed by transplant with HC or AD gut microbiota. Since the identification of the potential components driving microglial activation needs a large number of microglia cultures, we utilized rat but not mouse primary microglia cultures. We assessed microglial activation by analyzing its cellular morphology including diameter, branch points, and length. It was worth noting that the serum from the mice inoculated with AD-FMT significantly activated microglia with C/EBPβ transgenic mice stronger than WT mice. In contrast, the serum from Abx-treated WT or transgenic mice without any FMT treatment exhibited no activity (Supplementary Fig. 4A, B). To explore whether the microglial stimulatory activity was exerted by proteins or small molecules in the serum, we denatured and precipitated proteins with TCA, followed by high-speed centrifugation. Markedly, the supernatant after protein removal displayed comparable activity as the intact serum, suggesting that some small bioactive molecules are presumably responsible for this effect (Supplementary Fig. 5A, B). Next, we wondered whether the brain of the animals inoculated with AD-FMT also possess the same stimulatory activity on microglia, we employed the brain lysates and found the similar activity pattern as the serum (Supplementary Fig. 6A, B). Again, depleting proteins from the brain lysates prepared from the animals inoculated with AD-FMT did not affect the stimulatory effect on microglia activation, indicating that the bioactive small molecules might be responsible for the activity (Supplementary Fig. 7A, B). We also validated these effects with the brain lysates or serum from C/EBPβ Tg mice, inoculated with AD or HC-FMT, on primary microglia cultures from C/EBPβ Tg mice. As expected, mouse primary microglia

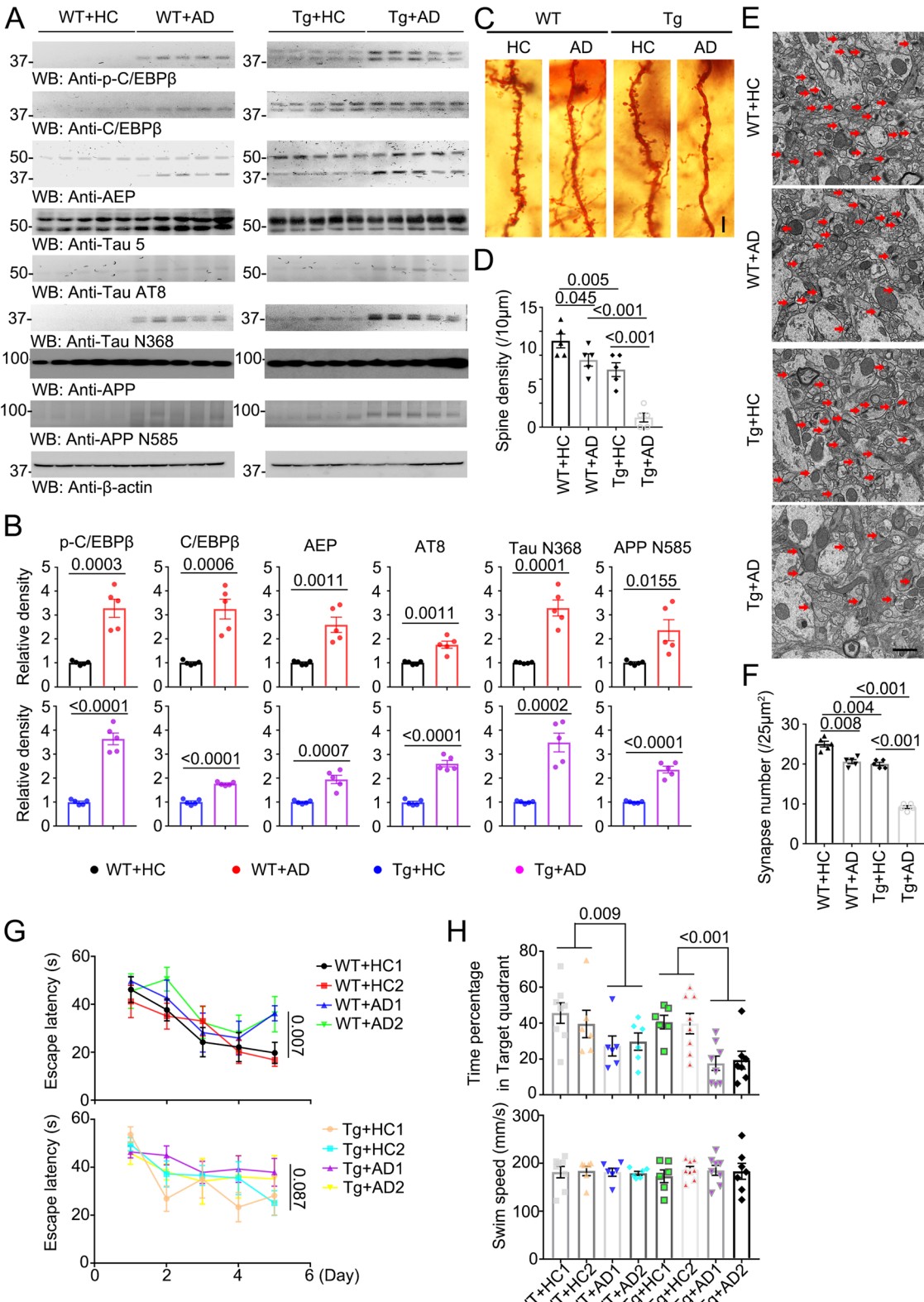

from C/EBPβ Tg mice were prominently activated by the brain lysates or serum from AD-FMT-treated mice (Supplementary Fig. 8).

To identify the molecules accounting for the microglial activation, we fractionated the supernatant from the cortex tissues from humanized Abx-treated C/EBPβ transgenic mice using HPLC and analyzed the microglia stimulatory activity using different fractions. Quantitative analysis indicated that fractions #15-#24 exhibited prominent activities from the brains inoculated with AD-FMT with the climaxed

effect in fractions #16-#17; in contrast, no demonstrable activity was observed in the same fractions from the brains treated with HC-FMT. Metabolomics analysis showed that PGE2 was mostly abundant in these two fractions (Supplementary Fig. 9A, B), suggesting that this PUFA metabolite might be accountable for the potent microglia stimulatory activity. Next, we quantitatively analyzed the concentrations of several major PUFA metabolites including AA, LTB4, 12-HHTrE, and PGE2 in the different tissues from the cortex, serum to the feces.

**Fig. 2 | AD humanized Abx-treated mice display increased toxic proteins and declining cognitive functions compared with HC humanized Abx-treated mice in WT and C/EBPβ Tg mice. A** Immunoblot showing p-C/EBPβ, C/EBPβ, AEP, APP and tau expression and processing in 6-month-old mouse brains of AD or HC humanized Abx-treated mice. **B** Quantitative analysis of immunoblot. The bands of p-C/EBPβ, C/EBPβ, AEP, AT8, TauN368, and APPN585 were measured with image J and normalized with β-actin. (n = 5 biologically independent samples in each group, data are shown as mean ± SEM, two-tailed Student's t test). **C** The dendritic spines from the apical dendritic layer of the hippocampus CA1 region were analyzed by Golgi staining. Scale bar: 10 μm. **D** Quantitative analysis of Golgi staining, (n = 5 biologically independent samples in each group, data are shown as mean ± SEM, one-way ANOVA and Bonferroni's multiple comparison test). **E** Electron microscopy analysis of synapses, the number of synapses in CA1 region of AD humanized

Abx-treated mice were decreased compared with HC humanized Abx-treated mice. Arrows indicate the synapses. Scale bar, 1 μm. **F** Quantitative analysis of synapses in CA1, (n = 5 biologically independent samples mice per group, mean ± SEM. one-way ANOVA and Bonferroni's multiple comparison test). (**G & H**) Morris Water Maze analysis of cognitive functions. AD humanized Abx-treated mice exacerbated the learning and memory dysfunctions, but does not affect motor function. (mean ± SEM., one-way ANOVA and Bonferroni's multiple comparison test). WT + HC1 (n = 8 biologically independent samples); WT + HC2 (n = 6 biologically independent samples); WT + AD1 (n = 6 biologically independent samples); WT + AD2 (n = 6 biologically independent samples); Tg+HC1 (n = 6 biologically independent samples); Tg +HC2 (n = 8 biologically independent samples); Tg +AD1 (n = 8 biologically independent samples); Tg +AD2 (n = 7 biologically independent samples).

Interestingly, AA, 12-HHTrE, and PGE2 but not LTB4 were selectively elevated in the cortex from both WT and C/EBPβ transgenic mice treated by AD-FMT with the latter much higher than the former (Supplementary Fig. 9C), and we observed similar but slightly different distribution patterns between the serum and feces samples (Supplementary Fig. 9D, E). To further confirm the effects of 12-HHTrE and PGE2 on the activation of microglia and Aβ42 deposition, we included these two metabolites to in vitro cultured C/EBPβ Tg microglia and neurons, and found that 12-HHTrE and PGE2 significantly augmented the activation of microglia and elevated neuronal Aβ42 deposition (Supplementary Fig. 10). To interrogate the roles of C/EBPβ in mediating PGE2 metabolite's effect, we knocked it down with shRNA from primary neurons, microglia or their mixture and monitored Tau phosphorylation or neuronal apoptosis upon PGE2 treatment. Depletion of C/EBPβ substantially diminished AT8 and TUNEL activities in neurons and CD86 signals in primary microglia. Hence, the deletion of C/EBPβ from primary neuronal cultures or microglia alleviated the detrimental effects of *B. Fragilis* metabolite (Supplementary Fig. 11). Biochemical analysis showed that neuron/microglia mixture displayed stronger AEP-mediated effects including APP N585, Tau N368 fragmentation, and Aβ and AT8 activities elevation by PGE2 than primary neurons alone, correlated with inflammatory cytokines escalation (Supplementary Fig. 12A, B). Neuronal apoptosis exhibited the similar pattern, indicating that PGE2 triggered more robust AD pathological effects in neurons in the presence of microglia (Supplementary Fig. 12C, D).

### *Bacteroides fragilis* is enriched in AD fecal transplanted Abx-treated Thy1-C/EBPβ mice

Gut dysbiosis in AD patients stimulates C/EBPβ/AEP pathway, triggering microglia activation and AA-associated inflammation[11]. To compare the human-derived gut microbiomes in Abx-treated C/EBPβ transgenic mice and AD patient fecal samples, we collected the fecal pellets from humanized C/EBPβ transgenic mice inoculated with two HC and two AD-FMT. Bacterial DNA was extracted, and 16S rRNA sequence analysis was performed. The microbiome analysis showed that human gut microbiota by FMT was successfully established in Abx-treated mice. At the phyla level, the taxonomic study revealed that the microbial community of FMT-recipient mice was similar to the microbial phyla of the human donor inoculum (Supplementary Fig. 13A). In agreement with our recent study with germ-free 3xTg mice, the recipient mice successfully harbored 83.3% of the genus-level taxa, which was found in the donor inoculum. Moreover, the beta diversity analysis principal coordinate plot (PCoA) indicated that the microbiota of the FMT-recipient mice was clustered together with their human donor counterparts (Supplementary Fig. 13B). Notably, our study confirmed that there were no significant alterations in the alpha diversity between the donor and FMT recipient mice. Therefore, these results support the successful engraftment of human gut microbiota in the Abx-treated mice. In addition, we found significant augmentation in the mean relative abundances of *B. fragilis, B. uniformis*, and *C. innocuum*, while a

significant decrease in the mean relative abundances of *B. ovatus* (Supplementary Fig. 13C). In our previous study, we found that three strains of bacteria (*B. intestinalis*, *B. fragilis*, and *B. xylanivsolvens*) are escalated AD humanized ex-germ-free mice[11]. *B. fragilis* was the only strain that was increased in two independent studies, thus, we chose to focus on this bacterial strain. Remarkably, quantitative analysis of the concentrations of AA and its metabolites in the culture medium from the *B. fragilis* demonstrated that these bacteria were indeed implicated in AA metabolism and were responsible for AA metabolites-related pathologies in Abx-treated mice (Supplementary Fig. 13D, E).

Thy-1 is a neuronal promoter that drives gene expression in the brain and the gut (enteric neurons). To test whether inoculation of AD patients' feces affects the enteric neurons that express C/EBPβ in the Thy-1-C/EBPβ mouse, we performed p-C/EBPβ/Aβ42 and p-C/EBPβ/AT8 co-IF on the enteric neurons in the gut sections and found that the enteric neurons were also influenced by the AD patients' feces as CNS neurons in the brain (Supplementary Fig 14). Next, we conducted the metabolomics and analyzed the lipid metabolism profiles for the feces and brain samples with respect to AD compared to HC humanized Abx-treated C/EBPβ transgenic mice. For the feces samples, the long-chain saturated fatty acids including myristate, palmitate, stearate and arachidate, long-chain monounsaturated fatty acid, and most of PUFA (except docosatrienoate) are all significantly increased from 1.5 to 3.5 folds (Supplementary Fig. 15A). For the brain tissues, we analyzed the metabolomic profiles for both C/EBPβ transgenic mice versus WT littermates on the baseline. For most of the fatty acids, the ratios were around 1, except for a couple of PUFA including docosatrienoate and dihimo-linoleate revealing 1.75 and 1.5, respectively, suggesting that neuronal overexpression do not drastically alter the lipid metabolism profiles in Thy1- C/EBPβ transgenic mice as compared to WT controls. However, the metabolomic profiles for the brains appeared significantly different with respect to AD compared to HC in humanized Abx-treated C/EBPβ transgenic mice. We found that the long-chain saturated fatty acids, long-chain monounsaturated fatty acid, and most of PUFA (except docosatrienoate and arachidonate) metabolites were significantly higher in the brain by FMT from AD compared to the HC-FMT group (Supplementary Fig. 15B), supporting that AD gut microbiota indeed pronouncedly change the long chain fatty acid, especially, PUFA metabolism. Therefore, *Bacteroides fragilis* is selectively augmented in humanized Abx-C/EBPβ transgenic mice upon AD-FMT, and this bacteria-mediated PUFA metabolism is escalated in the animals.

### Live *Bacteroides fragilis* induces AD-like pathologies in Thy1-C/EBPβ transgenic mice

Identification of the molecules accounting for microglia activation from the serum and brain of humanized Abx-C/EBPβ mice by AD-FMT, combined with microbiome analysis and metabolomic assay, indicates that *Bacteroides fragilis* might play an important role in mediating microglia activation and AA-associated inflammation in the brain of Thy1-C/EBPβ transgenic mice (Fig. 3). To ascertain that *Bacteroides*

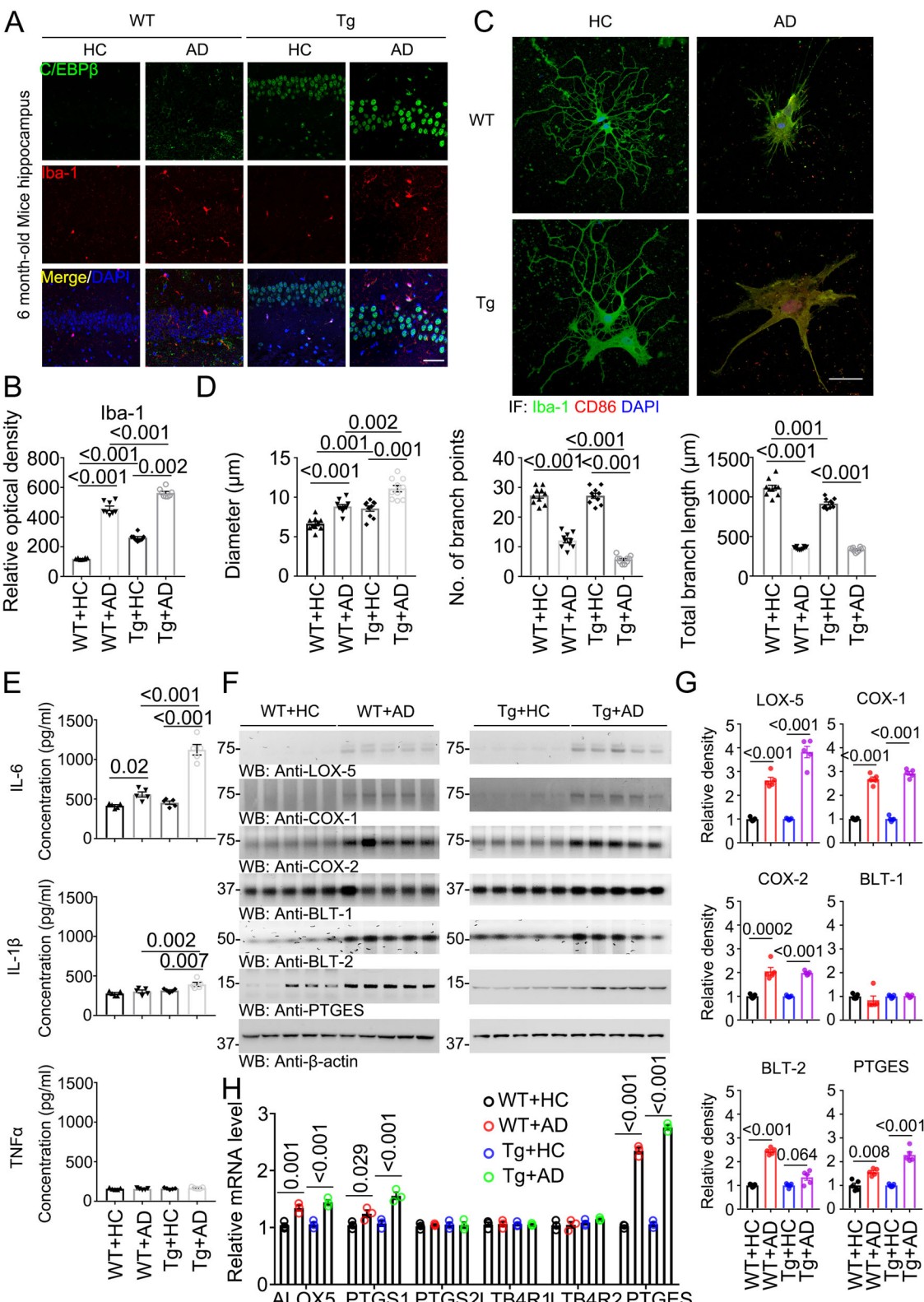

*fragilis* is indeed accountable for these pathological events, we inoculated dead or live *Bacteroides fragilis* into the Abx-pretreated Thy1-C/EBPβ transgenic mice via oral gavage. In two months, we performed IF co-staining and found that live but not dead *Bacteroides fragilis* or control medium stimulated aggregated Aβ in the cortex, which was ThS positive as well (Fig. 4A, C). Aggregated Tau T22 staining also validated that live but not dead *Bacteroides fragilis* elicited ThS-positive Tau accumulation in the hippocampal neurons (Fig. 4B, D).

Quantification revealed that mouse Aβ$_{42}$ levels in the brain were significantly elevated by live but not dead *Bacteroides fragilis*, while mouse Aβ$_{40}$ concentrations remained unchanged regardless of what treatment (Fig. 4E). IB analysis showed that live *Bacteroides fragilis* strongly induced Aβ production and aggregation as compared to medium or dead bacteria-treated samples. Remarkably, T22 immunoblotting showed aggregated p-Tau was demonstrable in the mice after live but not dead *Bacteroides fragilis* treatment (Fig. 4F). Silver

**Fig. 3 | AD humanized Abx-treated mice display increased activated microglia and inflammatory cytokines compared with HC humanized Abx-treated mice in WT and C/EBPβ Tg mice. A** Immunofluorescent staining of Iba-1 (red) and C/EBPβ (green) in the hippocampus CA1 region of the 6-month-old mice brains from AD or HC humanized Abx-treated mice. Scale bar: 40 μm. **B** Quantitative analysis of Iba-1 optical density (n = 8 biologically independent samples in each group, data are shown as mean ± SEM, one-way ANOVA and Bonferroni's multiple comparison test). **C** The high magnification of immunofluorescent staining of Iba-1 (green) and CD86 (red) positive microglia in the hippocampus CA1 region of the brains from AD or HC humanized Abx-treated mice. Scale bar: 40 μm. **D** Quantitative analysis of diameter, number of branch points, and total branch length in Iba-1 (green) positive microglia. (n = 8 biologically independent samples in each group, data are shown as mean ± SEM, one-way ANOVA and Bonferroni's multiple comparison test).

**E** Proinflammatory cytokine IL-1β, IL-6 and TNFα concentrations in the brain lysates from AD or HC humanized Abx-treated mice, respectively. (n = 5 biologically independent samples in each group, data are shown as mean ± SEM, one-way ANOVA and Bonferroni's multiple comparison test). **F** Immunoblot showing arachidonic acid metabolism products LOX-5, COX-1, COX-2, BLT-1, BLT-2, and PTGES expression and processing in mouse brains of AD or HC humanized Abx-treated mice. **G** Quantitative analysis of immunoblot. The bands of LOX-5, COX-1, COX-2, BLT-1, BLT-2, and PTGES were measured with image J and normalized with β-actin. (n = 5 biologically independent samples in each group, data are shown as mean ± SEM, two-tailed Student's t test). **H** Quantitative RT-PCR analysis of the brain samples from AD or HC humanized Abx-treated mice. Comparing C/EBPβ targeted arachidonic acid pathway genes. (n = 3 in each group, data are shown as mean ± SEM, two-tailed Student's t test).

staining confirmed that extensive protein inclusions were conspicuously increased in the cortex and the hippocampus from live but not dead *Bacteroides fragilis*-treated mice, indicating widespread pathological Aβ and Tau accumulations in Thy1-C/EBPβ transgenic mice (Fig. 4G, H). Golgi staining disclosed robust dendritic spine decrease after live but not dead *Bacteroides fragilis* treatment (Fig. 4I). EM analysis also exhibited the similar synapse reduction pattern by live *Bacteroides fragilis* (Fig. 4J, K). In alignment with the massive synaptic degeneration, cognitive behavioral tests demonstrated that live *Bacteroides fragilis* treatment significantly abrogated the learning and memory functions of Thy1-C/EBPβ transgenic mice (Fig. 4L, M). Hence, *Bacteroides fragilis* alone is sufficient to provoke AD-like pathologies and cognitive defects in Thy1-C/EBPβ transgenic mice.

### *Bacteroides fragilis* activates microglia and augments PUFA metabolites

AD gut microbiota activates C/EBPβ/AEP signaling in germ-free 3xTg mice[11] and Abx-treated C/EBPβ transgenic mice (Figs. 2 and 3). As expected, IF co-staining demonstrated that live *Bacteroides fragilis* treatment prominently upregulated C/EBPβ expression in Iba-1-positive microglia in the hippocampus as compared to dead bacteria or medium control (Fig. 5A, B). Iba-1 and CD86 co-staining validated that live *Bacteroides fragilis* strongly activated microglia as showed by the augmented diameter of the nucleus, decreased branch points, and lengths of the microglial processes when compared with medium and dead bacteria (Fig. 5C, D). Consequently, pro-inflammatory cytokines including IL-6 and IL-1β levels were significantly escalated by live but not dead *Bacteroides fragilis*, though TNFα concentrations were not different among the groups (Fig. 5E). In vitro quantification study disclosed that *Bacteroides fragilis* swiftly metabolized AA into a variety of PUFA derivatives (Supplementary Fig. 13D, E). Accordingly, in vivo quantitative analysis with bacteria-inoculated mouse brains showed that live but not dead *Bacteroides fragilis* potently augmented AA, 12-HHTrE, and PGE2 concentrations, whereas LTB4 levels remained unchanged (Fig. 5F). Thus, oral administration of *Bacteroides fragilis* strongly escalates C/EBPβ and activates microglia in Thy1-C/EBPβ transgenic mice, associated with augmented proinflammatory cytokines and AA-derived 12-HHTrE and PGE2 metabolites.

### 12-HHTrE or PGE2 treatment induces AD-like pathologies and cognitive defects in Thy1-C/EBPβ transgenic mice

The inflammatory cyclooxygenase-PGE2 pathway is implicated in the pre-clinical development of AD, both in the epidemiology of normal aging populations and in transgenic mouse models of Familial AD[40]. Both 12-HHTrE and PGE2 are metabolized from PGH2, which is derived from AA by COX-1/2. To explore whether these metabolites from PUFA by *Bacteroides fragilis* induce AD-like pathologies, we treated Thy1-C/EBPβ transgenic mice via i.p. (intraperitoneal) injection of the diluted 12-HHTrE, or PGE2 solution at the dose of 5 mg/kg twice a week for 4 weeks. IF co-staining showed that both compounds elicited apparently Aβ aggregated signals in the cortex, which were ThS positive with

PGE2 much stronger than 12-HHTrE (Fig. 6A, C). T22 staining also validated that both metabolites provoked p-Tau aggregates in the hippocampus. Again, PGE2 displayed much more robust activity than 12-HHTrE (Fig. 6B, D). Consistent with Aβ IF staining activities, ELISA quantification revealed that PGE2 significantly induced mouse Aβ_{42} production as compared to 12-HHTrE, which also displayed a much stronger effect than vehicle control. By contrast, Aβ_{40} concentrations remained constant, no matter whether the mice were treated with which compound or vehicle (Fig. 6E). IB analysis with anti-Aβ 4G8 echoed mouse Aβ_{42} levels by ELISA with PGE2 exhibiting stronger effect than 12-HHTrE. Anti-T22 displayed the similar pattern with PGE2 more robust than 12-HHTrE in triggering p-Tau aggregation (Fig. 6F). In agreement with the IB results, silver staining showed that both PGE2 and 12-HHTrE triggered abundant protein inclusions in the cortex and the hippocampus with the latter stronger than the former (Fig. 6G, H). Golgi staining and EM studies supported that both metabolites significantly elicited synaptic degeneration with PGE2 stronger than 12-HHTrE (Fig. 6I–K). Accordingly, MWM tests indicated that both metabolites significantly impaired cognitive functions with PGE2 more potent than 12-HHTrE, fitting with their pathological effects, although none of them affected the swimming speed (Fig. 6L, M). Thus, PGE2 exhibits a stronger effect in provoking AD-like pathologies and cognitive defects than 12-HHTrE in Thy1-C/EBPβ transgenic mice.

### 12-HHTrE or PGE2 treatment elicits microglia activation and elevates PUFA levels in the brain of Thy1-C/EBPβ transgenic mice

To interrogate whether these metabolites mimic AD-FMT or *Bacteroides fragilis* in activating C/EBPβ in microglia and microglia activation, we conducted IF co-staining and validated that Iba-1 positive microglia cells in the hippocampus from Thy1-C/EBPβ transgenic mice were highly increased upon 12-HHTrE or PGE2 stimulation, accompanied with escalation of C/EBPβ (Fig. 7A, B). Iba-1 and CD86 co-staining revealed that both compounds strongly agitated microglia, which were confirmed by quantitative analysis of the diameters, branch points, and branch length (Fig. 7C, D). As a result, these proinflammatory PUFA metabolites significantly increased both IL-6 and IL-1β, with PGE2 stronger than 12-HHTrE, though TNFα levels remained constant (Fig. 7E). Correspondingly, PGE2 levels in the brain were the highest in mice treated with PGE2, 12-HHTrE concentrations were the highest in mice treated with the same compounds. Noticeably, AA levels were significantly elevated by both metabolites, whereas LTB4 quantities remained comparable among the groups (Fig. 7F). To explore whether depletion of microglia confers protection in the AD inoculated mice, we fed Thy1-C/EBPβ Tg mice with PLX3397 (CSF1R Inhibitor)[41] to delete microglia, followed by PGE2 treatment, and found that deletion of microglia from Thy1-C/EBPβ Tg mice attenuated PGE2-induced AD pathologies (Supplementary Fig. 16). Together, these findings support that *Bacteroides fragilis* abundant in AD patient gut microbiomes may activate C/EBPβ/AEP pathway and stimulate microglia activation via PGE2 and 12-HHTrE, two major metabolites from PUFA by these bacteria.

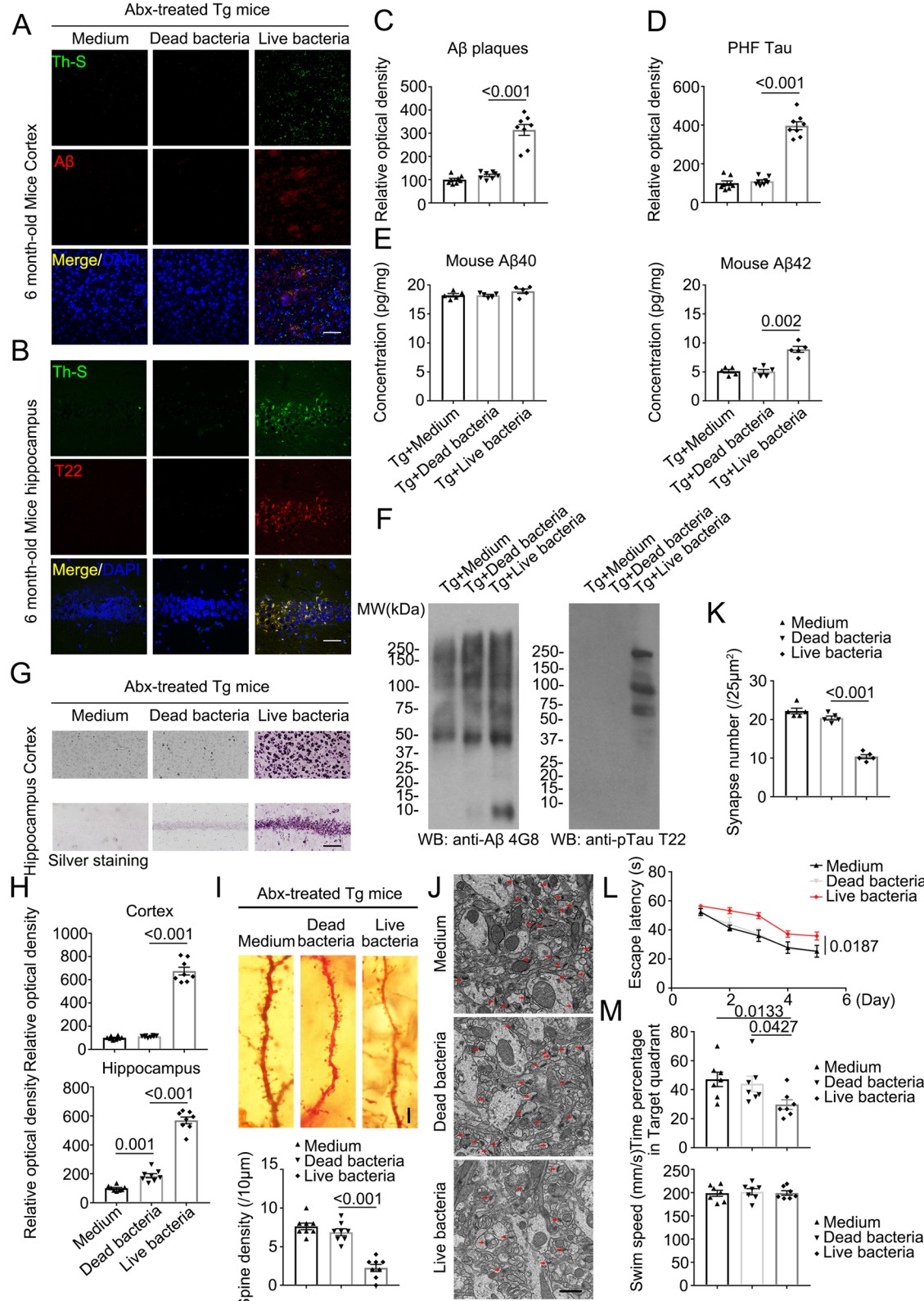

## Discussion

In the current work, we discover that *Bacteroides fragilis* bacteria is enriched in AD gut microbiota, which might be accountable for triggering AD-like pathologies associated with obvious microglia activation in Thy1-C/EBPβ transgenic mice, leading to cognitive dysfunctions. Depending on the analysis method, SPF mice usually harbor between 400 and 1000 bacterial strains in their gut

microbiota[42]. Employing microbiome analysis from Abx-treated mice inoculated with AD-FMT, we identify that *Bacteroides fragilis* and several other strains were strongly augmented. Interestingly, the same strain was also identified in our recent study with germ-free 3xTg AD mice inoculated with the AD gut microbiota[11]. Moreover, *Bacteroides* in the human subject's cohort where the AD samples originated, not only for this study but also in our prior report[11], were identified in higher

**Fig. 4 | Live bacteria treated mice display increased AD pathologies and impaired cognitive functions compared with control mice in C/EBPβ Tg mice.**
**A, B** Immunofluorescent staining of Aβ (red) in the frontal cortex region of brains, and T22 (red) in the hippocampus CA1 region of brains from 6-month-old Abx-treated C/EBPβ Tg mice after medium, dead bacteria, or live bacteria (*Bacteroides fragilis*) gavage for 2 months. Scale bar: 40 μm. **C, D** Quantitative analysis of positive plaques and PHFs (left panel, *n* = 8 biologically independent samples/group, data are shown as mean ± SEM, one-way ANOVA and Bonferroni's multiple comparison test). **E** Mouse Aβ$_{40}$ and Aβ$_{42}$ ELISA. The concentration of Aβ$_{42}$, not Aβ$_{40}$ was significantly increased in *Bacteroides fragilis* gavage Abx-treated C/EBPβ Tg mice compared with medium, or dead bacteria gavage Abx-treated C/EBPβ Tg mice cortex. (n = 5 biologically independent samples/group, data are shown as mean ± SEM, one-way ANOVA and Bonferroni's multiple comparison test). **F** Sarkosyl-insoluble fractions were blotted with antibodies against Aβ (4G8), or Tau T22 for Tau oligomers (representative of 3 mice). **G** Silver staining in cortex and hippocampus. Scale bar: 50 μm. **H** Quantitative analysis of G (n = 8 biologically independent samples/group, data are shown as mean ± SEM, one-way ANOVA and Bonferroni's multiple comparison test). **I** The dendritic spines from the apical dendritic layer of the hippocampus CA1 region were analyzed by Golgi staining. (Upper panel, Scale bar: 10 μm). Quantitative analysis of Golgi staining (n = 8 biologically independent samples/group, data are shown as mean ± SEM, one-way ANOVA and Bonferroni's multiple comparison test) (lower panel). **J** Electron microscopy analysis of synapses. Arrows indicate the synapses. Scale bar, 1 μm. **K** Quantitative analysis of synapses in CA1, (n = 5 biologically independent samples/group, mean ± SEM, one-way ANOVA and Bonferroni's multiple comparison test). **L, M** Morris Water Maze analysis of cognitive functions. Abx-treated C/EBPβ Tg mice after live bacteria (*Bacteroides fragilis*) gavage exacerbated the learning and memory dysfunctions, but did not affect motor function. (mean ± SEM.; *n* = 7 biologically independent samples/group, one-way ANOVA and Bonferroni's multiple comparison test).

abundance among AD patients with *Bacteroides fragilis*, in particular, being a top strain of AD among the cohort examined[10]. These findings therefore suggest that *Bacteroides fragilis* might implicate in triggering C/EBPβ/AEP pathway and microglia activation, resulting in demonstrable aggregated Aβ and Tau pathologies and extensive neuroinflammation.

Microglia play a crucial role in the clearance of these toxic protein assemblies[43]. However, with the progression of disease, notably in AD models, the healthy phagocytic responses of microglia to Aβ peptides fade away, either because microglia become ineffective or because they are overwhelmed by levels of accumulating Aβ peptides. In parallel with the progression of pathology in AD model mice, microglia also develop a more toxic inflammatory phenotype[44], leading to a damaging feed-forward cycle, with increasing buildup of toxic Aβ peptide assemblies along with escalated toxic cytokines. The pronounced microglia activation in both germ-free 3xTg mice and Abx-treated C/EBPβ transgenic mice upon AD-FMT called our attention in searching for the bioactive metabolites responsible for microglia stimulatory effect (Supplementary Figs. 4–8). Employing primary rat microglia activation assay, we identified that AA-derived two metabolites 12-HHTrE and PGE2 from the serum or brain samples from Abx-treated mice inoculated with AD-FMT were tightly coupled with microglia stimulatory activities (Supplementary Fig. 9). Interestingly, *Bacteroides fragilis* swiftly metabolized AA into various PUFA metabolites including 12-HHTrE and PGE2 (Supplementary Fig. 13D, E). Metabolomic analysis, combined with the HPLC study, demonstrated that PUFA metabolism was significantly elevated in the brain and feces from the mice after AD-FMT inoculation (Supplementary Fig. 16). Administration of these bioactive compounds into Thy1-C/EBPβ transgenic mice essentially recapitulated what we observed with AD-FMT or *Bacteroides fragilis* alone, underscoring that AD gut microbiota or *Bacteroides fragilis*-elicited AD-like pathologies and cognitive dysfunctions might be exerted via PGE2 or 12-HHTrE, which stimulated microglia activation and extensive neuroinflammation (Figs. 6 and 7).

Activated microglia exhibit diverse phenotypes and have multifaceted interactions with Aβ and tau species as well as with neuronal circuits. Active microglia exert various influences on the progression of AD, depending on the stage of disease, individual susceptibility, and state of microglial priming. Microglia could potentially be modulated at various points in the AD trajectory to either prevent or modify disease progression[45]. Highly ramified microglia can change to an ameboid form on pathological stimulation[46,47]. Microglia in aging brains have reduced branching that reduces their area of surveillance, which might contribute to the impairment of homeostatic functions[48–50]. Microglia in Braak stage V−VI brains display more profound morphological changes than do at earlier stages[51]. The temporal diversity of microglial morphology could be attributed to the intensity and duration of the pathological environment[52] but could also be related to the differential responses of microglia to different stimuli such as Aβ or Tau aggregation[53]. We show that the activated microglia, validated by Iba-1 and CD86 co-staining, reveal short processes in the brain with extensive aggregated Aβ and Tau pathologies in Thy1-C/EBPβ transgenic mice after AD-FMT, *Bacteroides fragilis* or PUFA metabolites treatment (Figs. 3, 5 and 7).

In addition to promoting the production of inflammatory mediators, C/EBP family members are themselves induced by the classical pro-inflammatory triad of IL-1β, IL-6, and TNFα[54–56], all of which are significantly increased in pathologically impacted regions of the AD brain[1]. It remains unclear why TNFα levels were stable in these animals regardless of the stimulation. Under AD-FMT treatment, we observed evident C/EBPβ/AEP signaling activation, accompanied by discernable APP N585 and Tau N368 fragmentation (Fig. 2A, B), which were in alignment with elevated mouse Aβ$_{42}$ and aggregated Aβ and Tau (Fig. 1). C/EBPβ regulates numerous gene expression in neuroinflammation[17,37]. Accordingly, C/EBPβ levels are highly increased in Iba-1 positive microglia cells in the brains of mice inoculated with AD-FMT, or treatment with live *Bacteroides fragilis* or 12-HHTrE/PGE2 (Figs. 3A, 5A and 7A). The activated C/EBPβ is further validated by IB with p-C/EBPβ and downstream target AEP (Fig. 2A, B). Noticeably, the mRNA transcription of the major enzymes implicating in AA metabolism including *ALOX5*, *PTGS1*, and *PTGES* (Fig. 3H). These findings are consistent with previous reports that the expression of COXs and LOX-5 and PTGES are mediated by C/EBPβ[37–39,57]. The augmented enzymes upregulate various AA metabolites including PGE2 and 12-HHT etc., which feedback to stimulate microglia activation, further escalating neuroinflammation. In alignment with 12-HHTrE augmentation, its receptor BLT2 levels are evidently escalated in both WT and C/EBPβ transgenic mice by AD-FMT.

The release of pro-inflammatory molecules can lead to synaptic dysfunction, neuronal death, and inhibition of neurogenesis[58]. For instance, IL-1β induces synaptic loss by increasing prostaglandin E2 (PGE2) production, which leads to presynaptic glutamate release and postsynaptic *N*-methyl-D-aspartate (NMDA) receptor activation[59]. In addition, the complement system is activated, promoting the phagocytic function of microglia, which might result in inappropriate pruning of synapses[60]. We observed that IL-1β was significantly elevated in the brain of Thy1-C/EBPβ transgenic mice, when they were treated with AD-FMT, live *Bacteroides fragilis*, or 12-HHTrE/PGE2 (Figs. 3E, 5E and 7E). In consequence, the dendritic spines and synapses in these animals were severely degenerated, resulting in substantial cognitive deficits (Figs. 2F, G; 4L, M, 6L, M). On the other hand, IL-6, a major downstream target of C/EBPβ, is also highly augmented, mirroring the IL-1β oscillation.

COX-1 and COX-2, which are capable of converting AA to prostaglandin H2 (PGH2), show a differential cell type-specific expression in the adult mammalian brain. COX-1 is constitutively expressed in most tissues and in the brain predominantly in microglia[26]. On the other hand, in the brain, COX-2 is constitutively expressed in

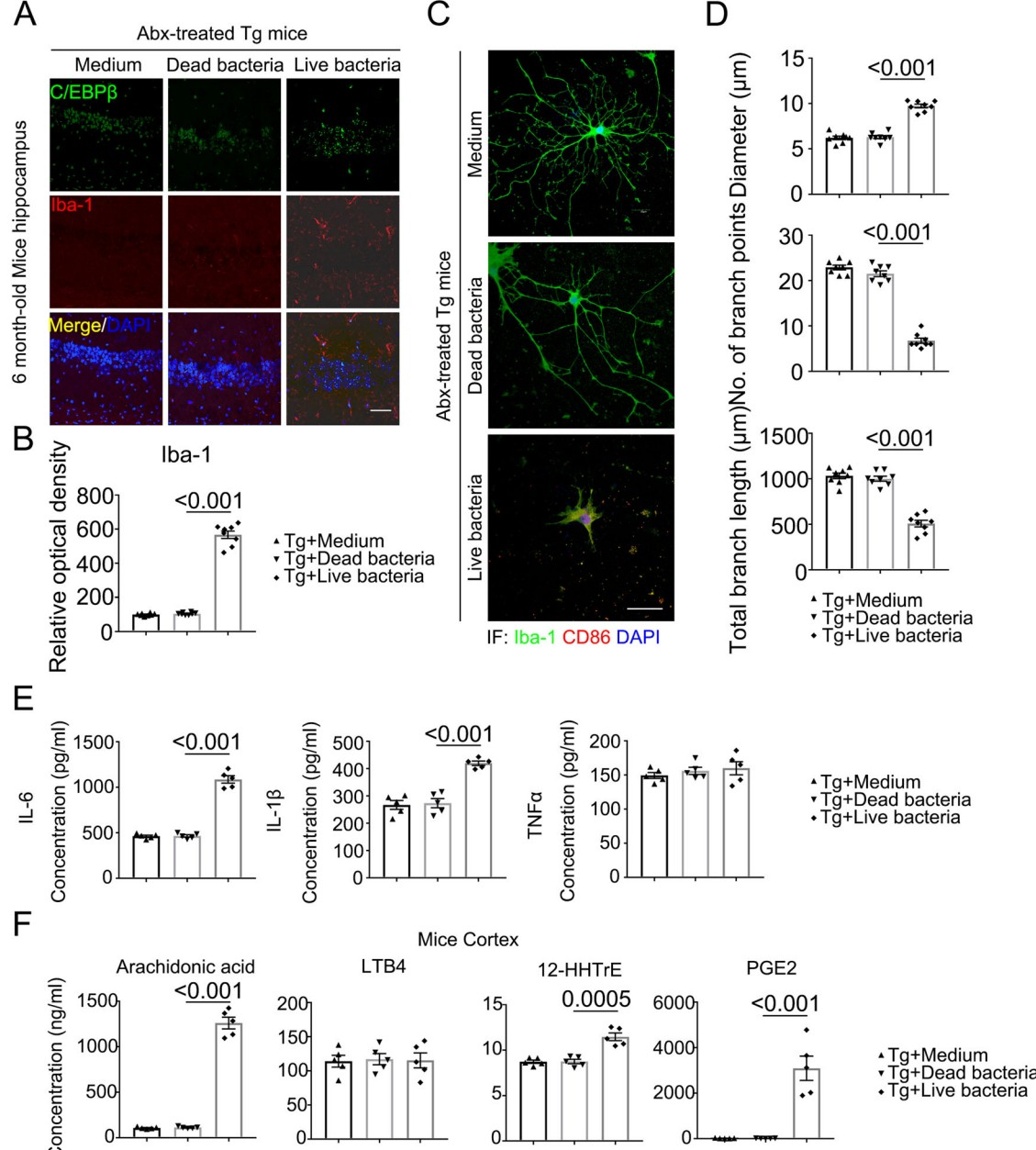

**Fig. 5 | Live bacteria treated mice display increased activated microglia and inflammatory cytokines compared with control mice in C/EBPβ Tg mice.**
**A** Immunofluorescent staining of Iba-1 (red) and C/EBPβ (green) in the hippocampus CA1 region of the brains from 6-month-old Abx-treated C/EBPβ Tg mice after medium, dead bacteria, or live bacteria (*Bacteroides fragilis*) gavage for 2 months. Scale bar: 40 μm. **B** Quantitative analysis of Iba-1 optical density (n = 8 biologically independent samples in each group, data are shown as mean ± SEM, one-way ANOVA and Bonferroni's multiple comparison test). **C** The high magnification of immunofluorescent staining of Iba-1 (green) and CD86 (red) positive microglia in the hippocampus CA1 region of the brains from Abx-treated C/EBPβ Tg mice after medium, dead bacteria, or live bacteria (*Bacteroides fragilis*) gavage for 2 months. Scale bar: 40 μm. **D** Quantitative analysis of diameter, number of branch

points, and total branch length in Iba-1 (green) positive microglia. (n = 8 biologically independent samples in each group, data are shown as mean ± SEM, one-way ANOVA and Bonferroni's multiple comparison test). **E** proinflammatory cytokine IL-1β, IL- 6 and TNFα concentrations in the brain lysates from Abx-treated C/EBPβ Tg mice after medium, dead bacteria, or live bacteria (*Bacteroides fragilis*) gavage for 2 months, respectively. (n = 5 biologically independent samples in each group, data are shown as mean ± SEM, one-way ANOVA and Bonferroni's multiple comparison test). **F** Concentrations of arachidonic acid (AA) and its metabolites in Abx-treated C/EBPβ Tg mice after medium, dead bacteria, or live bacteria (*Bacteroides fragilis*) gavage cortex. (n = 5 biologically independent samples in each group, data are shown as mean ± SEM, one-way ANOVA and Bonferroni's multiple comparison test).

hippocampal neurons and their dendritic spines. Neuronal COX-2 expression is modified by synaptic activity[61] as well as by pathological conditions including the Aβ-triggered neurotoxicity[62]. Depending upon the differential cellular localization of COX-1 and COX-2, the subsequent conversion of PGH2 into other active molecules may lead to dissimilar ultimate products of COX-1 vs. COX-2 activity. The cyclooxygenase-PGE$_2$ pathway modulates the inflammatory response

to accumulating Aβ peptides through actions of specific E-prostanoid G-protein coupled receptors. PGE$_2$ binds four G-protein coupled receptors (GPCRs) termed E-prostanoid receptors (EP1-4) that have distinct downstream signaling cascades and cellular distributions in the brain. In vivo, all four EP receptors are expressed in neurons; microglial expression of EP2, EP3, and EP4 receptors is validated in mouse brain[63–65]. Phenotypic changes in microglia, including

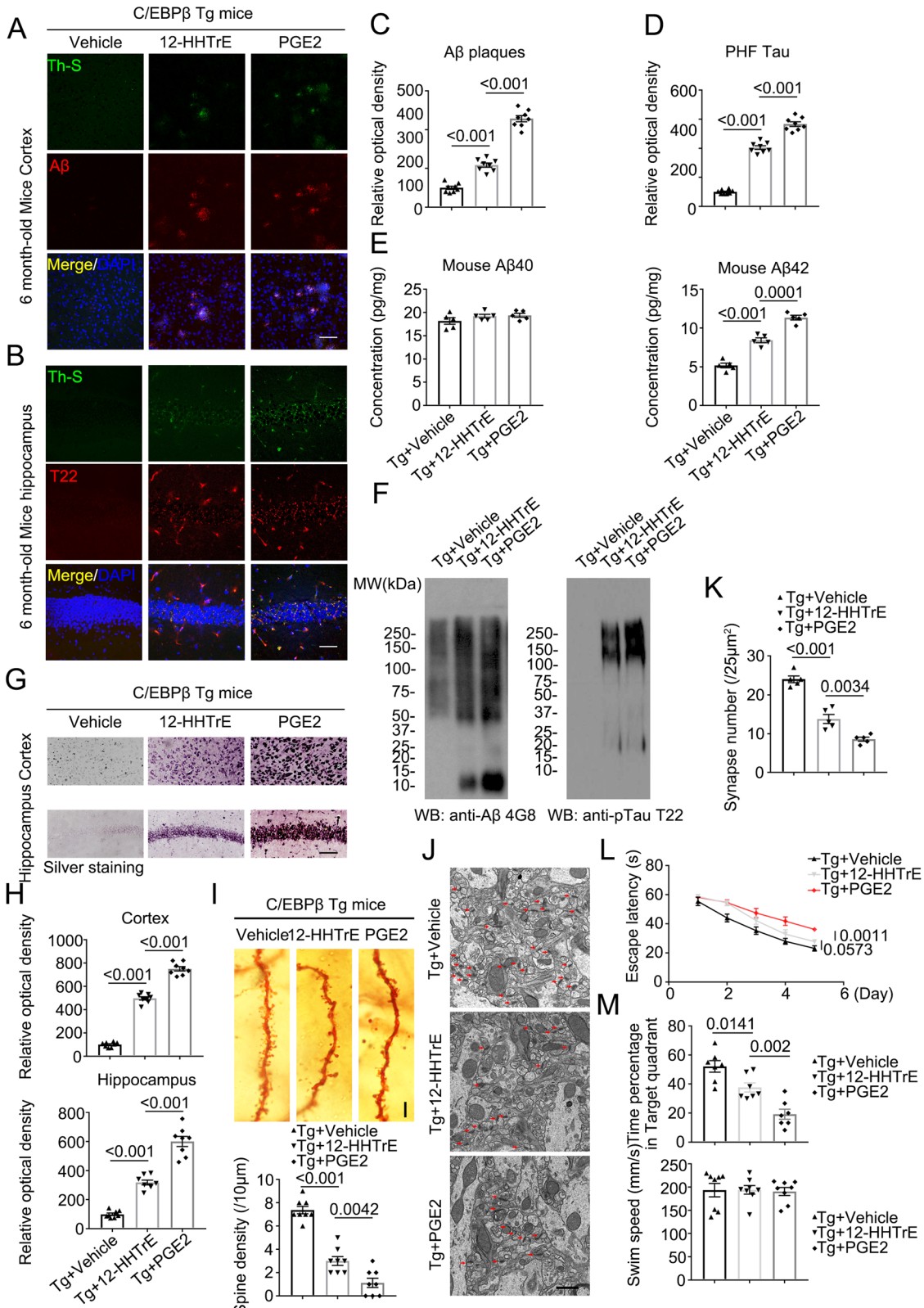

alterations in morphology, proteomic signatures, and behavior, are associated with disease progression[66]. Employing rat primary microglia morphology assay, we identified two COX-1/2 metabolites 12-HHTrE and PGE2 from the serum or brain of Abx-treated mice inoculated with AD-FMT (Supplementary Fig. 9), which exhibit robust microglia activation in the brain (Fig. 3C, D). The receptors by which microglia recognize Aβ species are mostly pattern recognition receptors, including Toll-like receptors, CD14, CD47, α6β1 integrin, and

scavenger receptors (including CD36). In turn, these receptors activate molecular pathways that induce microglial phenotypic changes[67–69]. Conceivably, the EP receptors on microglia cells might mediate these two metabolites' pathological effects including microglia morphology in vitro and in mouse brains.

Epidemiological studies support the possibility that anti-inflammatory drugs delay the onset and possible progression of AD. Anti-inflammatory treatment, including NSAIDs and steroids, might

**Fig. 6 | 12-HHTrE and PGE2 treated mice display increased AD pathologies and impaired cognitive functions in C/EBPβ Tg mice. A, B** Immunofluorescent staining of Aβ (red) in the frontal cortex region of brains, and T22 (red) in the hippocampus CA1 region of brains from 6-month-old vehicle, 12-HHTrE, or PGE2-treated C/EBPβ Tg mice. Mice received IP injection of the diluted 12-HHTrE, or PGE2 solution at the dose of 5 mg/kg twice a week for 4 weeks. Scale bar: 40 µm. **C, D** Quantitative analysis of positive plaques and PHFs (left panel, n = 8 biologically independent samples in each group, data are shown as mean ± SEM, one-way ANOVA and Bonferroni's multiple comparison test). **E** Mouse Aβ$_{40}$ and Aβ$_{42}$ ELISA. (n = 5 biologically independent samples in each group, data are shown as mean ± SEM, one-way ANOVA and Bonferroni's multiple comparison test). **F** Sarkosyl-insoluble fractions were blotted with antibodies against Aβ (4G8), or Tau T22 for Tau oligomers (representative of 3 mice). **G** Silver staining in cortex and hippocampus. Scale bar: 50 µm. **H** Quantitative analysis of (**G**). (n = 8 biologically independent samples in each group, data are shown as mean ± SEM, one-way ANOVA

and Bonferroni's multiple comparison test). **I** The dendritic spines from the apical dendritic layer of the hippocampus CA1 region were analyzed by Golgi staining. (upper panel, Scale bar: 10 µm). Quantitative analysis of Golgi staining (n = 8 biologically independent samples in each group, data are shown as mean ± SEM, one-way ANOVA and Bonferroni's multiple comparison test) (lower panel). **J** Electron microscopy analysis of synapses, the number of synapses in CA1 region of 12-HHTrE, or PGE2-treated C/EBPβ Tg mice were decreased compared with vehicle-treated mice. Arrows indicate the synapses. Scale bar, 1 µm. **K** Quantitative analysis of synapses in CA1, (n = 5 biologically independent samples per group, mean ± SEM, one-way ANOVA and Bonferroni's multiple comparison test). **L, M** Morris Water Maze analysis of cognitive functions. 12-HHTrE, or PGE2-treated C/EBPβ Tg mice exacerbated the learning and memory dysfunctions, but does not affect motor function. (mean ± SEM.; n = 7 biologically independent samples per group, one-way ANOVA and Bonferroni's multiple comparison test).

decrease the risk of AD by as much as 50%[70]. Nevertheless, inhibition of COX enzymatic activity by NSAIDs (non-steroidal anti-inflammatory drugs) has different consequences depending on the timing of AD development, and inhibition of COX-1/COX-2 by non-selective NSAIDs is beneficial in preventing disease in healthy aging individuals but ineffectual once symptoms begin[71,72]. In epidemiologic studies of cognitively normal aging populations, NSAIDs prevent and delay the development of AD[73–75]. Together, our study supports that gut dysbiosis in AD patients, especially *B. fragilis* enrichment and it's PUFA metabolites PGE2 and 12-HHTrE abundance, triggers C/EBPβ/AEP signaling activation, leading to AD-like pathologies in Thy1-C/EBPβ transgenic mice. Imaginably, pretreatment with NSAID on the prodromal AD *B. fragilis* carriers may significantly prevent or slow down AD onset and progression.

## Methods

### Ethical approval
The mice experimental protocol was reviewed and approved by the Institutional Animal Care and Use Committee (IACUC) at Emory University (DAR3000226ELMNTS-N). For the human fecal samples, we have received the written informed consent from the participants for the use of samples and data for this present research. This study was approved by the UMass Chan Medical School Institutional Review Board (IRB) at the University of Massachusetts Medical School (docket H00010892).

### Animals
For the generation of Thy1-human C/EBPβ transgenic mouse of C57BL/6J background, mouse genomic fragments containing homology arms (HAs) were amplified from bacterial artificial chromosome (BAC) clone by using high fidelity Taq, and were sequentially assembled into a targeting vector together with recombination sites and selection markers. After confirming correctly targeted ES clones via Southern Blotting, we selected some clones for blastocyst microinjection, followed by founder production. Founders were confirmed as germline-transmitted via crossbreeding with wild-type. In the end, male and female F1 heterozygous mutant mice were confirmed as the final deliverables for this project. The C57BL/6J mice were purchased from Jax lab (Cat#: 000664). These mice were free access to sterile water and food with 5 mice/ cage. They were kept in a specific pathogen-free (SPF) condition with 20–25 °C temperature and 45–55% humidity on a regular 12-h light/dark cycle.

### Abx gavage
The antibiotic cocktail was comprised of four antibiotics and one antifungal (ampicillin (100 mg/kg), vancomycin (50 mg/kg), metronidazole (100 mg/kg), neomycin (100 mg/kg), amphotericin B (1 mg/kg)). This protocol was maintained for 30 days. This cocktail was made fresh every 36 h. Previous studies show that this method of antibiotic administration is the most effective in depleting the microbiome with minimal effects on the morbidity and mortality of the animal[76].

### Primary cultured neuron and microglia
All pregnant C/EBPβ Tg mice are self-bred, all pregnant rats were purchased from the Jackson Laboratory. The protocol was reviewed and approved by the Emory Institutional Animal Care and Use Committee. Primary mouse cortical neurons, were isolated from E18 mice. Brain tissues were dissected, dissociated, and incubated with 5 ml of D-Hanks containing 0.25% trypsin for 5 min, centrifuged at 1000 g for 5 min after the addition of 4 ml of the neuronal plating medium containing Dulbecco's modified Eagle's medium/F12 with 10% fetal bovine serum. Then, the cells were resuspended, about 5 × 10$^5$ cells were plated onto each well of 12-well plates for WB, and 1 × 10$^5$ cells were plated onto each glass coverslip for cell imaging. The neurons were then put into a humidified incubator with 5% CO2 at 37 °C. The medium was changed to neurobasal medium supplemented with 2% B27 (maintenance medium) after 2 to 4 h. Neurons at 11 to 13 DIV were treated with drugs. For primary rat microglia, briefly, coat two T-75 culture flasks with 7 ml each of 10 µg/ml PDL for 2 h. Wash the flask bottom with distilled water 3 times before use. Tissues were dissected, dissociated, and incubated with 5 ml of D-Hanks containing 0.25% trypsin for 10 min, centrifuged at 1000 × g for 5 min after. Aspirate the supernatant and resuspend the pellet. Then plate each tube of cells into one coated T-75 flask at the density of 50,000 cells/cm². Add culture media (10% FBS/DMEM + 1% Pen/Strep) to reach a volume of 15 ml in the flasks. Put seeded flasks into an incubator with 5% CO$_2$, 100% humidity at 37 °C. Change culture media every 5 days. To collect microglia, vigorously tap the flasks on the bench top and collect the floating cells in conditioned culture media. Use a hemocytometer to count the floating cell density and seed the cells at 50,000 cells/cm² in PDL-coated culture vessels. The microglial is ready to use the next day.

### Human donor and criteria
The Frozen fecal samples obtained from 4 donors (2 AD and 2 HC (healthy control) were generously provided by Dr. John P. Haran and Dr. Beth A. McCormick. The human feces samples were collected from nursing home elders who are 65 years of age and lived in one of four nursing home facilities located in central Massachusetts. All the elders had been living at that facility for 1 month and did not have any diarrheal illness or antimicrobial exposure within the preceding 4 weeks. No elders suffered from dysphagia or had a feeding tube. Any elders with antimicrobial exposure or diarrheal illness during the conduct of the study were excluded from this analysis. Human donor selection criteria were as described previously[10].

### 12-HHTrE and PGE2 treatment
12-HHTrE and PGE2 (Cayman chemicals, 10 mg/ml) were brought to room temperature right before being used and diluted with sterilized 1xPBS to 1 mg/ml. Thy1-C/EBPβ Tg mice received IP injection of the diluted 12-HHTrE or PGE2 solution immediately at the dose of 5 mg/kg twice a week for 4 weeks.

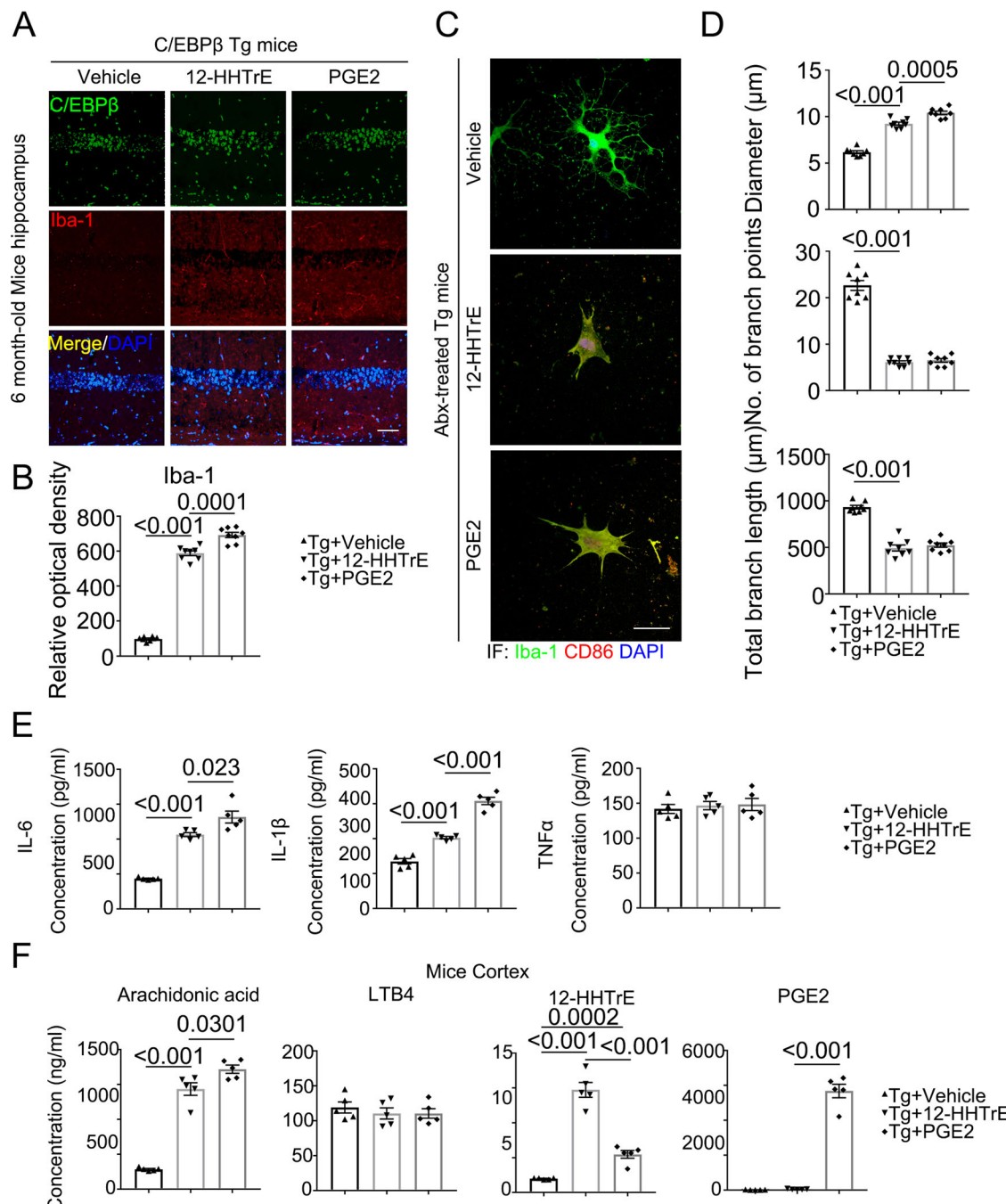

**Fig. 7 | 12-HHTrE and PGE2 treated mice display increased activated microglia and inflammatory cytokines in C/EBPβ Tg mice. A** Immunofluorescent staining of Iba-1 (red) and C/EBPβ (green) in the hippocampus CA1 region of the brains from 6-month-old vehicle, 12-HHTrE, or PGE2-treated C/EBPβ Tg mice. Scale bar: 40 μm. **B** Quantitative analysis of Iba-1 optical density (n = 8 biologically independent samples in each group, data are shown as mean ± SEM, one-way ANOVA and Bonferroni's multiple comparison test). **C** The high magnification of immunofluorescent staining of Iba-1 (green) and CD86 (red) positive microglia in the hippocampus CA1 region of the brains from vehicle, 12-HHTrE, or PGE2-treated C/EBPβ Tg mice. Scale bar: 40 μm. **D** Quantitative analysis of diameter, number of

branch points, and total branch length in Iba-1 (green) positive microglia. (n = 8 biologically independent samples in each group, data are shown as mean ± SEM, one-way ANOVA and Bonferroni's multiple comparison test). **E** Proinflammatory cytokine IL-1β, IL-6 and TNFα concentrations in the brain lysates from vehicle, 12-HHTrE, or PGE2-treated C/EBPβ Tg mice, respectively. (n = 5 biologically independent samples in each group, data are shown as mean ± SEM, one-way ANOVA and Bonferroni's multiple comparison test). **F** Concentrations of arachidonic acid (AA) and its metabolites in vehicle, 12-HHTrE, or PGE2-treated C/EBPβ Tg mice cortex. (n = 5 biologically independent samples in each group, data are shown as mean ± SEM, one-way ANOVA and Bonferroni's multiple comparison test).

## Immuno-staining

Treated microglia or cryo-sections of mouse brain were fixed with 4% PFA (Paraformaldehyde) for 10 min followed by 3 times wash in PBS, 10 min 0.5% Triton X-100 and 30 min blocking in 1% BSA, as well as the overnight incubation with Iba-1 antibody (VWR, 1:200) and C/ EBPβ antibody (H-7, 1:200) at 4 °C. To detect the localization of Aβ,

the slides were incubated with Aβ antibody (1:200) at 4 °C. To detect the localization of T22, the slides were incubated with T22 (1:500) at 4 °C. After overnight incubation, the slides were washed three times in PBS and incubated with Texas Red Red-conjugated anti-rabbit IgG or FITC-conjugated anti-mouse IgG for 1 h at room temperature or 0.0125% Thioflavin-S in 50% ethanol for 5 min. The slides were

washed three times in PBS, then covered with a glass cover using a mounting solution, and analyzed under a fluorescence microscope (Olympus).

## Golgi staining

Mice brains were fixed in 10% formalin for 24 h, and then immersed in 3% potassium bichromate for 3 days in the dark. The solution was changed each day. Then the brains were transferred into 2% silver nitrate solution and incubated for 24 h in the dark. Vibratome sections were cut at 60 μm, air dried for 10 min, dehydrated through 95% and 100% ethanol, cleared in xylene and cover-slipped. For the measurement of spine density, only spines that emerged perpendicular to the dendritic shaft were counted.

## Gallyas silver staining

Silver staining was performed using the Gallyas method. After deparaffinization, 5-μm sections were incubated in 5% periodic acid for 5 min, washed in water, and then placed in an alkaline silver iodide solution (containing 1% silver nitrate) for 1 min. The sections were then washed in 0.5% acetic acid for 10 min, and placed in developer solution for 15 min, before washing with 0.5% acetic acid, then water. The sections were then treated with 0.1% gold chloride for 5 min before washing in water, and incubated in 1% sodium thiosulphate (hypo) for 5 min, before a final wash[77].

## Real-time PCR

RNAs from tissues were isolated with Trizol. After reverse transcription with SuperScriptIII reverse transcriptase, Real-time PCR reactions were performed using the ABI 7500-Fast Real-Time PCR System, Gene-specific primers and TaqMan Universal Master Mix Kit were designed and bought from Taqman (please find the details in Table 1). All kits and reagents were purchased from Life Technologies. $2^{-\Delta\Delta Ct}$ method was used for the relative quantification of gene expression. For each data point, at least 2 duplicated wells were used.

## Detergent-insoluble protein aggregates preparation for Western blots

The soluble and insoluble proteins were extracted following protocol[78] and then were analyzed by Western blots.

## Western Blotting

The tissues were washed with ice-cold PBS and lysed in RIPA buffer (20 mM Tris-HCl, pH 7.5, 1 mM EDTA, 1 mM EGTA, 150 mM NaCl, 2.5 mM sodium pyrophosphate, 1% NP-40, 1% sodium deoxycholate, 1 mM $Na_3VO_4$ and 1 mM β-glycerophosphate) with protease inhibitor cocktail for 20 min on ice. The supernatant was collected by centrifuging at 14,000 rpm for 20 min at 4 °C. Then the protein extract was diluted to 5 mg/ml. After electrophoresis, the samples were incubated overnight at 4 °C with primary antibody, followed by immunoblotting. Please find the primary antibodies in Table 1. Uncropped and unprocessed scans of the blots were shown as Source Data.

## ELISA quantification

The tissues were diluted with cold reaction buffer (PBS, 5% BSA, 0.03% Tween-20, protease inhibitor cocktail), and centrifuged at 16,000 g for 20 min at 4 °C. Then analyzed with mouse $A\beta_{40/42}$, IL-6, TNFα, IL-1β, PGE2, and LTB4 ELISA kit according to the manufacturer's instructions.

## Proinflammatory cytokines ELISA

The mice brain tissue was lysed in lysis buffer (50 mM Tris, pH 7.4, 40 mM NaCl, 1 mM EDTA, 0.5% Triton X-100, 1.5 mM $Na_3VO_4$, 50 mM NaF, 10 mM sodium pyrophosphate, 10 mM sodium β-glycerophosphate, supplemented with protease inhibitors cocktail), and centrifuged for 15 min at

16,000 g. Then the supernatants were analyzed by IL-6, TNFα, and IL-1β ELISA kit according to the manufacturer's instructions. The proinflammatory cytokines concentrations were determined by comparison with the standard curve.

## Protocol for microbiota analysis

DNA was extracted from stool samples (n = 5 mice per group) using a PowerSoil kit from MO BIO Laboratories (Carlsbad, CA). 16 S rRNA genes were PCR-amplified from each sample using a composite forward primer and a reverse primer containing a unique 12-base barcode, designed using the Golay error-correcting scheme, which was used to tag PCR products from respective samples. We used primers for paired-end 16 S community sequencing on the Illumina platform using bacteria/archaeal primer 515 F/806 R. Primers were specific for the V4 region of the 16 S rRNA gene. The forward PCR primer sequence contained the sequence for the 5' Illumina adapter, the forward primer pad, the forward primer linker, and the forward primer sequence. Each reverse PCR primer sequence contained the reverse complement of the 3' Illumina adapter, the Golay barcode (each sequence contained a different barcode), the reverse primer pad, the reverse primer linker, and the reverse primer. Three independent PCR reactions were performed for each sample, combined, and purified with AMPure magnetic purification beads (Agencourt). The products were quantified, and a master DNA pool was generated from the purified products in equimolar ratios. The pooled products were sequenced using an Illumina MiSeq sequencing platform. Bioinformatics analysis was performed using QIIME2 (Quantitative Insights into Microbial Ecology, r2023.2). Sequences were assigned to OTUs with UPARSE using 97% pairwise identity and were classified taxonomically using the RDP classifier retrained with Greengenes. After chimera removal, the average number of reads per sample was 21,511. A single representative sequence for each OTU was aligned using PyNAST 1.0, and a phylogenetic tree was then built using FastTree. The phylogenetic tree was used to compute the UniFrac distances. The PCoA analysis shown is unweighted. The raw sequence data reported in this paper have been deposited in the DDBJ (accession number: SSUB021355).

## In vitro anaerobic bacteria culture experiment

*Bacteroides fragilis* (25285, ATCC) was cultured in the chopped meat medium (AS-811, AnaerobeSystems) and incubated at 37 °C in Bio-bag with the GasPak™ EZ Gas Generating Container Systems (260001, Gaspak EZ W/Indicator, BD). For the arachidonic acid metabolism assay, the above bacteria were grown in a medium added with or without 62.5 μM arachidonic acid. 72 h later, the medium from each culture was collected for the HPLC analysis or ELISA assay.

## HPLC quantitative analysis of arachidonic acid and its metabolites

Solid phase extraction (SPE) sample preparation: Samples were centrifuged for 15 min at 4000 rpm at 4 °C to remove bacteria and media pellets. 10 ml of the supernatant was collected and 10 μl of formic acid and 1 ml of methanol were added to bring the media to contain 10% methanol and 1% formic acid. The samples were sat on ice for at least 15 mins to allow protein precipitation. The samples were centrifuged for 15 min at 4000 rpm at 4 °C to remove any precipitations.

As for the control media without inoculating bacteria, the stock standards were added to bring the standard concentration to x50, X100 and X1000 of the stock concentration. The stock concentrations of PGE2, Arachidonic acid, LTB4 and 12HHTrE were 200 μg/ml. 250 μg/ml, 2 μg/ml and 2 μg/ml respectively. The compounds were extracted using SPE columns (Thermo Scientific, CA). Columns were washed with 2 ml of MeOH followed by 2 ml of $H_2O$, 10% methanol with 1% formic acid. 8 ml of the samples or 2 ml of the control medium was added for extraction. After applying the sample, the columns were washed with 4 ml of 10% MeOH, and the analytes were then eluted with 1 ml of

**Table 1 | Key resources for the experiments, including the antibodies, chemicals, primers, kits, animals, software, etc**

| Reagent or Resource | Source | Identifier |
|---|---|---|
| Antibodies | | |
| Anti-δ-secretase (AEP) (11B7) | Dr. Colin Watts (University of Dundee) | N/A, 1:1000 for WB, 1:200 for IF |
| Anti-TauN368 | Home-made | N/A, 1:1000 for WB, 1:200 for IF |
| Anti-APPN585 | Home-made | N/A, 1:1000 for WB, 1:200 for IF |
| Anti-C/EBPβ(H-7) HRP | Santa Cruze | Cat# sc-7962 HRP, 1:1000 for WB, 1:200 for IF |
| Anti-phoshpo Thr235 C/EBPβ | Cell Signaling Technology | Cat# 3084 s, 1:1000 for WB, 1:200 for IF |
| Anti-beta Amyloid 4G8 | Covance | Cat# SIG-39220, 1:1000 for WB, 1:200 for IF |
| Anti-Tau (T22) oligomeric | Millipore Sigma | Cat# ABN454, 1:1000 for WB, 1:500 for IF |
| Anti-phospho-Tau (Ser202, Thr205) (AT8) | Thermo-fisher | Cat# MN1020, 1:1000 for WB, 1:200 for IF |
| Anti-Iba1 | VWR | Cat# 100369-764, 1:1000 for WB, 1:200 for IF |
| Anti-CD86 (E2G8P) | Cell Signaling Technology | Cat# 91882 S, 1:1000 for WB, 1:200 for IF |
| Anti-Tau (TAU-5) | Invitrogen | Cat# AHB0042, 1:1000 for WB |
| Anti-PTGES | Thermo-fisher | Cat# PA5-60916, 1:1000 for WB |
| Anti-APP A4 (22C11) | Millipore Sigma | Cat# MAB348, 1:1000 for WB |
| Anti-5 Lipoxygenase (LOX5) | Abcam | Cat# ab39347, 1:1000 for WB |
| Anti-Cyclooxygenase 1 (COX1) | Abcam | Cat# ab109025, 1:1000 for WB |
| Anti-Cyclooxygenase 2 (COX2) | Abcam | Cat# ab15191, 1:1000 for WB |
| Anti-LTB4-R1/BLTR | Thermo-fisher | Cat# BS-2654R, 1:1000 for WB |
| Anti-LTB4-R2 | Thermo-fisher | Cat# BS-2655R, 1:1000 for WB |
| Anti-β actin (AC-15) | Invitrogen | Cat# AM4302, 1:1000 for WB |
| Anti-mouse IgG-HRP | Cell Signaling Technology | Cat# 7076 S, 1:2000 for WB |
| Anti-rabbit IgG-HRP | Cell Signaling Technology | Cat# 7074 S, 1:2000 for WB |
| Anti-mouse IgG-Alexafluor 488 | Invitrogen | Cat# A11001, 1:2000 for IF |
| Anti-mouse IgG-Alexafluor 594 | Invitrogen | Cat# A11005, 1:2000 for IF |
| Anti-rabbit IgG-Alexafluor 488 | Invitrogen | Cat# A11034, 1:2000 for IF |
| Anti-rabbit IgG-Alexafluor 594 | Invitrogen | Cat# A11037, 1:2000 for IF |
| Anti-rabbit IgG-Cy5 | Invitrogen | Cat# A10523. 1:2000 for IF |
| Chemical and antibiotics | | |
| Thioflavin S | Millipore Sigma | Cat# T1892-25G |
| Arachidonic acid | Cayman Chemical | Item No. 90010 |
| Prostaglandin E2 | Cayman Chemical | Item No. 14010 |
| Leukotriene B4 | Cayman Chemical | Item No. 20110 |
| 12-HTTrE | Cayman Chemical | Item No. 34590 |
| Vancomycin Hydrochloride | VWR | Cat# 97062-554 |
| Gentamicin sulfate | Sigma-Aldrich | Cat# G1914-5G |
| Ampicillin | Sigma-Aldrich | Cat# A159325 |
| Erythromycin | Sigma-Aldrich | Cat# PHR1039-1G |
| Neomycin | Fisher Scientific | Cat# 21810031 |
| Critical Commercial Assay | | |
| Amyloid beta 42 mouse ELISA Kit | Invitrogen | Cat# KMB3441 |
| Amyloid beta 40 mouse ELISA Kit | Invitrogen | Cat# KMB3481 |
| Mouse IL-6 ELISA Kit | Abcam | Cat# ab100712 |
| TNF alpha Mouse Uncoated ELISA Kit | Invitrogen | Cat# 88-7324-88 |
| IL-1 beta Mouse Uncoated ELISA Kit | Invitrogen | Cat# 88-7013-22 |
| Prostaglandin E2 (PGE2) ELISA Kit | caymanchemical | Cat# 514010 |
| Leukotriene B4 (LTB4) ELISA Kit | caymanchemical | Cat# 10009292 |
| Deposited Data | | |
| 16 s RNA sequences | PPMS for EIGC https://fex.cores.emory.edu/ | KYE22363-63838 |
| RNAseq data | PPMS for EIGC https://fex.cores.emory.edu/ | KYE22363-64287 |
| Experimental Models: Organisms/Strains | | |
| Mouse: C/EBPβ Tg | Ye lab | N/A |
| Bacteria: *Bacteroides fragilis* VPI 2553 | ATCC | Cat# 25285 |
| Sequence-Based Reagents | | |
| ALOX 5 TaqMan® Gene Expression Assays | Thermo-fisher | Cat# Mm01182747_m1 |

**Table 1 (continued) | Key resources for the experiments, including the antibodies, chemicals, primers, kits, animals, software, etc**

| Reagent or Resource | Source | Identifier |
|---|---|---|
| PTGS1 TaqMan® Gene Expression Assays | Thermo-fisher | Cat# Mm00477214_m1 |
| PTGS2 TaqMan® Gene Expression Assays | Thermo-fisher | Cat# Mm00478374_m1 |
| LTB4R TaqMan® Gene Expression Assays | Thermo-fisher | Cat# Mm00521839_m1 |
| LTB4R2 TaqMan® Gene Expression Assays | Thermo-fisher | Cat# Mm00498491_s1 |
| PTGES TaqMan® Gene Expression Assays | Thermo-fisher | Cat# Mm00452105_m1 |
| GAPDH TaqMan® Gene Expression Assays | Thermo-fisher | Cat# Mm99999915_g1 |
| Software and Algorithms | | |
| ImageJ 1.51J8 | National Institutes of Health | https://imagej.nih.gov/ij/ |
| SPSS 12.0 | IBM | https://www.ibm.com/analytics/spss-statistics-software |
| Prism-GraphPad 7.04 | GraphPad | https://www.graphpad.com/scientific-software/prism/ |
| QIIME 2 r2023.2 | Dr.Alam Ashfaqul laboratory, University of Kentucky | https://qimme.org/ |
| Illumina's bcl2fastq | Dr.Alam Ashfaqul laboratory, University of Kentucky | https://www.illumina.com/ |
| PyNAST 1.0 | Dr.Alam Ashfaqul laboratory, University of Kentucky | http://biocore.github.io/pynast/ |

MeOH. The eluent was diluted 1:1 with $H_2O$ to bring down the methanol to 50% for the HPLC assay. The injection volume was 200 μl and 20 μl.

Arachidonic acid and its metabolites, LTB4, 12HHTrE, and PGE2 were measured by high-performance liquid chromatography with a photodiode array autosampler, model 1525 binary pump, and model 2996 photodiode array detector was used. Analytes were separated using reverse-phase chromatography on a Waters Xbridge BEH column (4.6 × 150 mm, 5 μM, Waters) with a guard cartridge (3.9 × 5 mm, 5 μM, Waters). The analytes were identified by comparing their retention times and spectral profiles to known standards. A stock solution in 100% methanol and the concentrations are 250 μg/ml for arachidonic acid, 2 μg/ml of LTB4 and 12HHTrE, 200 μg/ml for PGE2, respectively. The working standard solution was prepared in 50% methanol. A series dilution of the standard was prepared in 50% methanol as the standard curve.

## Liquid chromatography coupled to mass spectrometry analysis of metabolites (LC-MS/MS)

Samples were dried and resuspended in 5% acetonitrile and 45% isopropanol followed by separation on Ascentis Express Capillary C18 column (2.7 μm beads, 15 cm × 300 μm; Supelco, Bellefonte, PA). The analytes were eluted over a 35-min gradient at a rate of 2 μl/min with buffer B ranging from 1% to 75% (buffer A: 10 mM ammonium acetate in 40% acetonitrile and 60% water; buffer B: 10 mM ammonium acetate in 90% isopropanol and 10% acetonitrile). The mass spectrometric analysis was performed on a Qexactive HF mass spectrometer (Thermo-Fisher Scientific, San Jose, CA) in negative mode. MS settings included MS1 scans (120,000 resolution, 100–1500 m/z, $3 × 10^6$ AGC and 50 ms maximal ion time) and 20 data-dependent MS2 scans (60,000 resolution, $2 × 10^5$ AGC, -100 ms maximal ion time, HCD, Stepped NCE (20, 40, 60), and 20 s dynamic exclusion). PGE2 was identified in samples and confirmed by comparing with the PGE2 standard with matched m/z, retention time and ms2 spectrum.

## Electron Microscopy

After deep anesthesia, mice were perfused transcardially with 2% glutaraldehyde and 3% paraformaldehyde in PBS. Hippocampal slices were post-fixed in cold 1% $OsO_4$ for 1 h, dehydrated, and flat-embedded in Araldite resin. Samples were prepared and examined using standard procedures. Ultrathin sections (90 nm) were stained with uranyl acetate and lead acetate and viewed at 100 kV in a JEOL 200CX electron microscope. Synapses were identified by the presence of synaptic vesicles and postsynaptic densities. Ten images were taken from each mouse to count synapses and the mean synapse count of each animal (5 mice per group) was used for statistical analysis.

## Immunocytochemistry in transmission electron microscopy

Following perfusion fixation, the brains were removed from the skull and blocks of tissue were cut from the stratum pyramidale of the CA1 sector of the mice hippocampus, incubated overnight in 4% paraformaldehyde in PBS, and rinsed three times (15 min each step) in PBS. Dehydrated in the ethanol gradient, then infiltrated with 100% ethanol/LR White resin (EMS, Hatfield PA) (25% ethanol/LR White, 50%, 75%, three changes of 100%) at 1 hr each step. The sample was left in the last change of LR White for 4 hrs and then placed in gelatin capsules for polymerization at 50 C for 12 h. The samples were sectioned on an RMC MTX ultra-microtome approx. 60 nm thick and placed on 200 mesh nickel grids. Each section incubated with antibodies was adjoining sections on each grid. For each brain sample, sections on grids were blocked in 3% BSA in PBS for 45 min at RT. They were then incubated in either Aβ mouse antibody (1:100 ab/diluent; 1% BSA in PBS) or AT8 mouse (1:100 ab/diluent) for 1 h at RT. Samples were washed in 1% BSA three times 10 min each step, then incubated in 1:20 GAM 15 nm gold (EMS, Hatfield PA) in diluent gold secondary antibody for 1 hr at RT. They were rinsed in filtered DI water 10 min each step for three times. Negative controls were exposed to secondary antibodies only. The sections were then viewed on a JEOL JEM1011 TEM at 100 kV JEOL USA, Peabody MA). Images were captured on an AMT mid-mount camera (AMT, Woburn MA).

## Statistical analysis

The sample size was determined by Power and Precision V2 (Biostat). Image J 1.51J8, SPSS 12.0, Prism-GraphPad 7.04, QIIME2 r2023.2, PyNAST 1.0, and Illumina's bcl2fastq were used for analysis. All data are expressed as Mean ± S.E.M. from three or more independent experiments, and the level of significance between the two groups was assessed with Student's t-test. For more than two groups, one-way ANOVA followed by LSD post hoc test was applied. A value of $p < 0.05$ was considered to be statistically significant.

## Reporting summary

Further information on research design is available in the Nature Portfolio Reporting Summary linked to this article.

# Data availability

Source data are provided with this paper. Source Data.xlsx includes the relevant raw data from each graphic. Supplementary Data 1.zip contains the metabolomic data of brain, serum, and fecal samples from the Alzheimer's disease mouse model, using proteomics only to quantify certain specific metabolites. Raw RNA sequencing data used in this study are available in the Bioproject database under accession codes PRJNA1001130 Source data are provided with this paper.

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

## Acknowledgements

This work is supported by a grant from the National Institute of Health (RO1, AG065177) to K. Ye. The Natural Science Foundation of Hubei Province of China for Distinguished Young Scholars (Grant No. 2022CFA104) to Y Xia. Additional support was provided by the Georgia Clinical & Translational Science Alliance of the National Institutes of Health under Award Number UL1TR002378, and National Institute Health (RO1 AG067483) to J.P. Haran. This study was supported in part by the Emory Gnotobiotic Animal (EGAC), which is subsidized by the Emory University School of Medicine and is one of the Emory Integrated Core Facilities. Additional support was provided by the Rodent Behavioral Core (RBC), which is subsidized by the Emory University School of Medicine and is one of the Emory Integrated Core Facilities; the Emory Integrated Genomics Core (EIGC), which is subsidized by the Emory University School of Medicine and is one of the Emory Integrated Core Facilities; as well as Emory HPLC Bioanalytical Core (EHBC), which was

supported by the Department of Pharmacology, Emory University School of Medicine. The metabolomics analysis on the feces and brain samples from the AD and HC humanized Abx-mice was performed by Metabolon, Inc. Morrisville, NC. We are thankful for Dr. Haiyan Tan at St. Jude Children's Research Hospital for identifying the chemicals in the HPLC fractions.

## Author contributions

K.Y., Y.Y.X conceived the project, designed the experiments, analyzed the data and wrote the manuscript. Y.Y.X., Y.F.X. designed and performed most of the experiments and analyzed the data. Z.-H.W. X.L. conducted genotype and breed the transgenic mice. J.P.H., B.A.M. provided the AD and HC fecal samples. A.M.A., X.J.S, X.C.W assisted with data analysis and interpretation and critically read the manuscript.

## Competing interests

The authors declare no competing interest.

## Additional information

[1]Department of Pathology and Laboratory Medicine, Emory University School of Medicine, Atlanta, GA 30322, USA. [2]School of Medicine, Jianghan University, Wuhan, HB 430056, China. [3]University of Kentucky, Microbiology, Immunology & Molecular Genetics Office - MN 376, Medical Science Building, 800 Rose Street, Lexington, KY, USA. [4]Department of Emergency Medicine, University of Massachusetts Chan Medical School, Worcester, MA, USA. [5]Microbiology and Physiological Systems, University of Massachusetts Chan Medical School, Worcester, MA, USA. [6]Program in Microbiome Dynamics, University of Massachusetts Chan Medical School, Worcester, MA, USA. [7]Department of Pathophysiology, School of Basic Medicine, Key Laboratory of Education Ministry of China/Hubei Province for Neurological Disorders, Tongji Medical College, Huazhong University of Science and Technology, Wuhan 430030, China. [8]Co-innovation Center of Neurodegeneration, Nantong University, Nantong, Jiangsu, China. [9]Faculty of Life and Health Sciences, Shenzhen Institute of Advanced Technology, Shenzhen Guangdong 518055, China. [10]These authors contributed equally: Yiyuan Xia, Yifan Xiao. ✉e-mail: xijishu@jhun.edu.cn; wxch@mails.tjmu.edu.cn; kq.ye@siat.ac.cn

