## [Peer Review File · Nature Communications]

REVIEWER COMMENTS

Reviewer #1 (Remarks to the Author):

The manuscript by Xia et al. reports a detrimental role of *Bacteroides Fragilis* in the pathogenesis of Alzheimer's disease (AD). The authors inoculate human fecal samples to C/EBP β Tg mice and observe more severe cognitive impairments, increased Ab deposition and up-regulated activation of microglia in Tg mice received AD patient fecal materials as compared to that received feces of HC. From the AD feces recipients, the authors further identify 12-HHTrE and PGE2 as the metabolites that exacerbates AD pathogenesis in C/EBP β Tg mice. *Bacteroides Fragilis* stands out by 16s analysis and were transplanted back to antibiotic-treated mice, where the 12-HHTrE and PGE2 increased consequently, leading to enhanced activation of microglia, elevated level of A β 42 deposition and impaired cognition. While the study provides solid evidence that gut microbes influence the manifestation and progression of AD, addressing the additional points below would greatly improve the prudence of the manuscript.

Major points:

1. The authors compared the gut microbial composition between AD patient feces and HC feces recipient mice and identified the enrichment of *B. Fragilis* in the Tg mice with AD patient feces. However, does *B. Fragilis* also show difference between AD patient and HC fecal samples that were used in this study?
2. In fig. 2G, AD microbiome were shown to have detrimental impact on WT mice instead of Tg mice. The authors claimed that C/EBP β overexpression has more dominant effect compared to AD microbiome. However, Tg mice with AD feces and WT mice with AD feces started and ended up with quite similar escape latency, which is contradictory to the authors' interpretation. The authors should clarify.
3. In fig. 3C, the authors showed that AD fecal samples are able to upregulate CD86 in representative microglia, indicating higher activation of microglia. It would be important to analyze CD86, as well as other inflammatory cytokines in the microglia in large quantities.
4. As CD86 serves as an activation marker for microglial phagocytosis. It would be also important to compare the phagocytotic ability of microglia from C/EBP β Tg mice with AD feces to other groups.
5. To further confirm the effects of 12-HHTrE and PGE2 on activation of microglia and A β 42 deposition, it would be important to supply these two metabolites to in vitro cultured Tg microglia and neurons.

Minor points:

1. In the introduction section, the author should also discuss the effects of gut microbiota on microglia in aging in addition to the neurodevelopment.
2. The background of IBA1 staining showed in fig. 3A is quite different across the groups, which weakens the conclusion. It would be more rigorous to select better representatives.
3. It would be important to explain the rationale of using rat microglia instead of Tg mouse microglia in the in vitro cell culture.

Reviewer #2 (Remarks to the Author):

In this study, Xia et al. identified *Bacteroides Fragilis* as the key strain to produce PUFA metabolites, which caused microglial activation and AD-associated neuropathology. The authors also demonstrated that *B. Fragilis* contributed to AD pathogenesis using different in vivo models. However, gut dysbiosis-induced the microglial activation, AD-related neuropathology and CREB/ β activation have been previously reported by the same group (Chen C, et al. *Gut* 2022;0:1–20. doi:10.1136/gutjnl-2021-326269). This study's innovation and impact need to be re-considered. Moreover, there are some concerns about the model systems used in this study. The authors claimed that the metabolites from *B. Fragilis* can induce microglial activation through the microglial CREB/ β signaling pathway. However, they used Thy-1-CREB/ β transgenic mice, and Thy-1 is a neuronal promoter. How could neuronal CREB/ β link to microglial activation? Is it directly or indirectly? My concerns in details are listed below:

1. Thy-1 is a neuronal promoter that drives the gene expression in the brain and the gut (enteric neurons). Does inoculation of AD patients' feces affect the enteric neurons that express CREB/ β in the Thy-1-CREB/ β mouse?
2. How is Thy-1-driven CREB expression in neurons related to microglial activation induced by *B. Fragilis* metabolites? Would conditional knock-out of CREB/ β in neurons or microglia rescue the detrimental effect of *B. Fragilis* metabolites?
3. Which one is more important in driving AD-related neuropathology, CREB/ β expressing neurons, or activated microglia? Would depletion of microglia confer protection in the AD inoculated mice?

4. The morphology of the microglia, as shown in the figures, is kind of strange/uncommon. Particularly in supplementary figures 2-5, why do the authors use rat primary microglia? Can they use mouse ones? There might be a cross-species effect. Also, the authors need to do staining of M1 markers to indicate the activation of microglia in vitro. The microglial morphology shown in Fig 5C and Fig 7C is so different.

5. To confirm that *B. Fragilis* is the key strain mediating AD pathogenesis, the authors need to inoculate the mouse with AD patients' feces in which the *B. Fragilis* is specifically depleted. This would exclude the effect of other bacterial strains.

EMORY
UNIVERSITY
SCHOOL OF
MEDICINE

Department of Pathology and Laboratory Medicine

Keqiang Ye, Ph.D.

Professor

Room 141, Whitehead Building

615 Michael Street

Atlanta, GA 30322

Telephone: (404) 712-2814 / Fax: (404) 712-2979

E-mail: kq.ye@siat.ac.cn

June 12th, 2023

We greatly appreciate the reviewers' favorable support of this work and their positive comments. We have carefully addressed all of the reviewers' concerns on a point-by-point basis with additional experimentation. The new data are included in 8 new Supplementary Figures, including Fig S2, Fig S3, S8, Fig S10-12; Fig S14, and Fig S16. The newly included text is highlighted in yellow. The specific response is listed below.

Reviewer #1 (Remarks to the Author):

The manuscript by Xia et al. reports a detrimental role of *Bacteroides Fragilis* in the pathogenesis of Alzheimer's disease (AD). The authors inoculate human fecal samples to C/EBP β Tg mice and observe more severe cognitive impairments, increased A β deposition and up-regulated activation of microglia in Tg mice received AD patient fecal materials as compared to that received feces of HC. From the AD feces recipients, the authors further identify 12-HHTrE and PGE2 as the metabolites that exacerbate AD pathogenesis in C/EBP β Tg mice. *Bacteroides Fragilis* stands out by 16s analysis and were transplanted back to antibiotic-treated mice, where the 12-HHTrE and PGE2 increased consequently, leading to enhanced activation of microglia, elevated level of A β 42 deposition and impaired cognition. While the study provides solid evidence that gut microbes influence the manifestation and progression of AD, addressing the additional points below would greatly improve the prudence of the manuscript.

Major points:

1. The authors compared the gut microbial composition between AD patient feces and HC feces recipient mice and identified the enrichment of *B. Fragilis* in the Tg mice with AD patient feces. However, does *B. Fragilis* also show difference between AD patient and HC fecal samples that were used in this study?

A: According to the previous report, *B. Fragilis* displays higher level in AD patients than HC fecal samples that were used in this study¹ (PMID: 31064831, Fig.3C). The same fecal samples, generously provided by Dr. John P. Haran and Dr. Beth A. McCormick, were employed in our previous work published in Gut (Chen et al., 2022). The information has been included in the upper part of Page 19.

2. In fig. 2G, AD microbiomes were shown to have detrimental impact on WT mice instead of Tg mice. The authors claimed that C/EBP β overexpression has more dominant effect compared to AD microbiome. However, Tg mice with AD feces and WT mice with AD feces started and ended up with quite similar escape latency, which is contradictory to the authors' interpretation. The authors should clarify.

A: In the MWM learning test, AD fecal samples exhibited significant difference in escape latency compared to HC samples in WT mice, but this effect was not obvious for Thy1-C/EBP β Tg mice (Fig 2G). However, AD fecal samples-treated WT and Tg mice spent much fewer time in the target quadrant than HC samples in the memory test, with Tg mice even shorter than WT mice, supporting that AD gut microbiota reveal stronger detrimental effect in cognitive dysfunctions in Thy1-C/EBP β mice than WT mice, which are in agreement with the pathological changes. Our previous studies show that Thy1-C/EBP β Tg mice display abnormal latency in MWM tests as compared to the different age groups or Thy1-ApoE4/C/EBP β Tg mice (Wang et al., 2022, Xia et al., 2022). This large baseline variation may smear the differences triggered by AD fecal samples, whereas AD gut dysbiosis exhibited noticeable effect in WT mice.

3. In fig. 3C, the authors showed that AD fecal samples are able to upregulate CD86 in representative microglia, indicating higher activation of microglia. It would be important to analyze CD86, as well as other inflammatory cytokines in the microglia in large quantities.

A: As required, we performed the co-IF in the microglia with other inflammatory cytokines (IL-1 β and IL-6). The CD86 and other inflammatory cytokines in active microglia cells are quantitatively analyzed (revised Fig.S2).

4. As CD86 serves as an activation marker for microglial phagocytosis. It would be also important to compare the phagocytotic ability of microglia from C/EBP β Tg mice with AD feces to other groups.

A: As required, we cultured microglia cells from WT or C/EBP β Tg mice treated with HC/AD fecal samples, then human A β ₄₂ aggregates were added into the medium. Forty-eight h later, remnant human A β ₄₂ aggregates were collected from the medium and analyzed with the A β ₄₂ ELISA kit. On the other hand, the incubated microglia cells were co-stained with anti-human A β ₄₂ and Iba-1. The phagocytotic ability of microglia from mice with AD feces was significantly enhanced (revised Fig S3).

5. To further confirm the effects of 12-HHTrE and PGE2 on activation of microglia and A β ₄₂ deposition, it would be important to supply these two metabolites to *in vitro* cultured Tg microglia and neurons.

A: As required, *in vitro* cultured microglia and neurons from Tg mice were employed for analysis of microglia activation and A β ₄₂ deposition. Indeed, 12-HHTrE and PGE2 augmented the activation of microglia and elevated neuronal A β ₄₂ deposition (revised Fig S10).

Minor points:

1. In the introduction section, the author should also discuss the effects of gut microbiota on microglia in aging in addition to the neurodevelopment.

A: As required, the effects of gut microbiota on microglia in aging has been added on page 3, line 10-13.

2. The background of IBA1 staining showed in fig. 3A is quite different across the groups, which weakens the conclusion. It would be more rigorous to select better representatives.

A: As required, higher quality graphs were replaced in revised fig. 3A.

3. It would be important to explain the rationale of using rat microglia instead of Tg mouse microglia in the *in vitro* cell culture.

A: This concern was also raised by Reviewer 2. The *in vitro* identification for active metabolites required a large number of microglia cells for activation screening, so we employed rat but not mouse embryos for primary microglia cultures. As required, Tg mouse microglia were used *in vitro* cell cultures in the revised manuscript (revised Fig S8).

Reviewer #2 (Remarks to the Author):

In this study, Xia et al. identified *Bacteroides Fragilis* as the key strain to produce PUFA metabolites, which caused microglial activation and AD-associated neuropathology. The authors also demonstrated that *B. Fragilis* contributed to AD pathogenesis using different *in vivo* models. However, gut dysbiosis-induced the microglial activation, AD-related neuropathology and CREB/ β activation have been previously reported by the same group (Chen C, et al. Gut 2022;0:1–20. doi:10.1136/gutjnl-2021-326269). This study's innovation and impact need to be re-considered. Moreover, there are some concerns about the model systems used in this study. The authors claimed that the metabolites from *B. Fragilis* can induce microglial activation through the microglial C/EBP β signaling pathway. However, they used Thy-1-C/EBP β transgenic mice, and Thy-1 is a neuronal promoter. How could neuronal C/EBP β link to microglial activation? Is it directly or indirectly? My concerns in details are listed below:

1. Thy-1 is a neuronal promoter that drives the gene expression in the brain and the gut (enteric neurons). Does inoculation of AD patients' feces affect the enteric neurons that express CREB/ β in the Thy-1-CREB/ β mouse?

A: The referee might mis-type the mouse strain name employed in the current study, the mice were Thy1-C/EBP β but not Thy1-CREB/ β mice. To address the raised question, we performed

p-C/EBP β /A β ₄₂ and C/EBP β /AT8 co-IF on the enteric neurons in the gut sections (revised Fig. S14), and the data showed that the enteric neurons were also influenced by the AD patients' feces/ *B. Fragilis*.

2. How is Thy-1-driven CEBP expression in neurons related to microglial activation induced by *B. Fragilis* metabolites? Would conditional knock-out of CREB/ β in neurons or microglia rescue the detrimental effect of *B. Fragilis* metabolites?

A: C/EBP β is age-dependently escalated in neurons, and we have reported that it increases the expression of *APP*, *MAPT*, *BACE1*, *ApoE* and *LGMN* mRNA expression. To explore the pathological roles neuronal C/EBP β in AD pathogenesis, we generated a neuronal specific C/EBP β transgenic mice. Our previous studies established that neuronal C/EBP β overexpression alone in the Thy1-C/EBP β Tg mice is not sufficient to trigger AD pathologies (Wang et al., 2022). However, in the context of neuronal ApoE4, we show that Thy1-ApoE4/C/EBP β Tg mice display AD pathogenesis, accompanied with extensive neuro-inflammation and activated microglia. Moreover, under high-fat diet (HFD) treatment, we also showed that Thy1-C/EBP β Tg mice develop AD pathologies, which were blunt by aspirin pretreatment (Liu et al., 2022), underscoring that neuro-inflammation and microglia activation are the prerequisites for neuronal C/EBP β -driven AD pathogenesis. Hence, these previous studies support that activation of microglia may be the crucial trigger for Thy-1-driven C/EBP β expression in neurons to demonstrate neuropathologies. Neuronal C/EBP β overexpression in Thy1-C/EBP β Tg mice results in inflammatory cytokines upregulation, because C/EBP β is a crucial transcription factor for various cytokines including IL-6 and IL-1 β etc., which robustly drive microglial activate in the brain.

Deletion of C/EBP β from primary neuronal cultures or microglia alleviate the detrimental effect of *B. Fragilis* metabolites (revised Fig S11). Furthermore, our previous studies show that heterozygous or homozygous knockout of C/EBP β or deleting neuronal C/EBP β via viral its specific shRNA substantially diminishes A β and Tau pathologies and microglia activation in 5xFAD, 3xTg or Tau P301S mice (Wang et al., 2018; Xia et al., 2020; Xiong et al., 2022), supporting that neuronal C/EBP β is indispensable for AD pathogenesis and neuroinflammation. Most importantly, C/EBP β is also a pivotal transcription factor for 12-HHTr and PGE2 metabolism-related genes (Gorgoni B et al., 2001, JBC; He D et al., 2008, Biochem J.; Straccia M et al., 2013, Glia), hence, deleting C/EBP β from either neurons or microglia cells may abolish PUFA-mediated neuroinflammation triggered by gut microbiota, observed in our most recent Gut paper (Chen et al., 2022), which will preclude us to rescue the detrimental effect of *B. Fragilis* metabolites.

3. Which one is more important in driving AD-related neuropathology, CEBP/ β expressing neurons, or activated microglia? Would depletion of microglia confer protection in the AD inoculated mice?

A: Our recent work shows that Thy-1-C/EBP/ β Tg mice fail to produce AD-like pathological changes by itself except under stimulations such as ApoE4 (Wang et al., 2022), 27-OH

cholesterol (Wang et al., 2021), or High-fat diet (Liu et al., 2022). Blocking neuro-inflammation and microglia activation by aspirin blunts HFD-induced AD pathogenesis in Thy1-C/EBP β mice, accompanied by reduced microglial activation (Liu et al., 2022). Hence, activated microglia is indispensable for neuronal C/EBP β overexpression to exhibit AD pathologies. Compared with neurons, mixed neurons and microglia cultures showed higher inflammatory cytokines, A β ₄₂, AT8 levels, and more apoptosis (revised Fig. S11 & S12). As requested, we fed Thy1-C/EBP β Tg mice with PLX3397 (CSF1R Inhibitor)² to delete microglia, followed by PGE2 treatment, we found that deletion of microglia attenuated PGE2-induced AD pathologies in Thy1-C/EBP β Tg mice (revised Fig. S16).

4. The morphology of the microglia, as shown in the figures, is kind of strange/uncommon. Particularly in supplementary figures 2-5, why do the authors use rat primary microglia? Can they use mouse ones? There might be a cross-species effect. Also, the authors need to do staining of M1 markers to indicate the activation of microglia *in vitro*. The microglial morphology shown in Fig 5C and Fig 7C is so different.

A: The *in vitro* screening process of metabolites required a large number of microglia cells, so we chose rats but not mice for primary microglial cultures. As requested, mouse primary microglia were also used in the *in vitro* cell cultures. CD86 (one of M1 markers) were used to indicate the activation of microglia *in vitro* (revised Fig. S8).

5. To confirm that *B. Fragilis* is the key strain mediating AD pathogenesis, the authors need to inoculate the mouse with AD patients' feces in which the *B. Fragilis* is specifically depleted. This would exclude the effect of other bacterial strains.

A: We appreciate the comment. However, there is no method available to specifically remove *B. fragilis* from AD fecal samples at present. Currently available techniques can remove a certain type of bacteria in the mixed flora *in vitro*, but most of them are aimed at *Escherichia coli*³⁻⁸. We will conduct the suggested experiments, once the specific bacteria strain depletion techniques are available in the near future. Nevertheless, treatment with the *B. fragilis* alone or even its inflammatory metabolite PGE2 in Thy1-C/EBP β Tg mice stimulates AD pathologies strongly support that *B. fragilis* may be the major contributor in AD gut microbiome to induce microglia activation and AD pathologies in Thy1-C/EBP β Tg mice.

Once again, thank you so much for monitoring our manuscript. I trust with these additional data, the revised manuscript is ready for publication! We look forward to hearing from you soon.

Best Regards,

Keqiang Ye, Ph.D.

Reference:

- 1 Haran, J. P. *et al.* Alzheimer's Disease Microbiome Is Associated with Dysregulation of the Anti-Inflammatory P-Glycoprotein Pathway. *mBio* **10**, doi:10.1128/mBio.00632-19 (2019).
- 2 Green, K. N., Crapser, J. D. & Hohsfield, L. A. To Kill a Microglia: A Case for CSF1R Inhibitors. *Trends Immunol* **41**, 771-784, doi:10.1016/j.it.2020.07.001 (2020).
- 3 Bikard, D. *et al.* Exploiting CRISPR-Cas nucleases to produce sequence-specific antimicrobials. *Nat Biotechnol* **32**, 1146-1150, doi:10.1038/nbt.3043 (2014).
- 4 Lam, K. N. *et al.* Phage-delivered CRISPR-Cas9 for strain-specific depletion and genomic deletions in the gut microbiome. *Cell Rep* **37**, 109930, doi:10.1016/j.celrep.2021.109930 (2021).
- 5 Hamilton, T. A. *et al.* Efficient inter-species conjugative transfer of a CRISPR nuclease for targeted bacterial killing. *Nat Commun* **10**, 4544, doi:10.1038/s41467-019-12448-3 (2019).
- 6 Ronda, C., Chen, S. P., Cabral, V., Yaung, S. J. & Wang, H. H. Metagenomic engineering of the mammalian gut microbiome in situ. *Nat Methods* **16**, 167-170, doi:10.1038/s41592-018-0301-y (2019).
- 7 Jin, W. B. *et al.* Genetic manipulation of gut microbes enables single-gene interrogation in a complex microbiome. *Cell* **185**, 547-562 e522, doi:10.1016/j.cell.2021.12.035 (2022).
- 8 Rubin, B. E. *et al.* Species- and site-specific genome editing in complex bacterial communities. *Nat Microbiol* **7**, 34-47, doi:10.1038/s41564-021-01014-7 (2022).

REVIEWERS' COMMENTS

Reviewer #1 (Remarks to the Author):

The authors perform great work to address my concerns. Thank you.

Reviewer #2 (Remarks to the Author):

The authors addressed the concerns of the reviewer. There is no further questions.